# Effects of surface geometry on light exposure, photoacclimation and photosynthetic energy acquisition in zooxanthellate corals

**Tomás López-Londoño**[1,2]*, **Susana Enríquez**[2], **Roberto Iglesias-Prieto**[1,2]

**1** Department of Biology, The Pennsylvania State University, University Park, Pennsylvania, United States of America, **2** Unidad Académica Puerto Morelos, Instituto de Ciencias del Mar y Limnología, Universidad Nacional Autónoma de México, Cancún, México

* tolopez@gmail.com

**Data Availability Statement:** All relevant data are within the paper and its Supporting Information files. Additionally, the data relevant to this study are available from figshare at https://doi.org/10.6084/m9.figshare.24599499.v1.

## Abstract

Symbiotic corals display a great array of morphologies, each of which has unique effects on light interception and the photosynthetic performance of *in hospite* zooxanthellae. Changes in light availability elicit photoacclimation responses to optimize the energy balances in primary producers, extensively documented for corals exposed to contrasting light regimes along depth gradients. Yet, response variation driven by coral colony geometry and its energetic implications on colonies with contrasting morphologies remain largely unknown. In this study, we assessed the effect of the inclination angle of coral surface on light availability, short- and long-term photoacclimation responses, and potential photosynthetic usable energy. Increasing surface inclination angle resulted in an order of magnitude reduction of light availability, following a linear relationship explained by the cosine law and relative changes in the direct and diffuse components of irradiance. The light gradient induced by surface geometry triggered photoacclimation responses comparable to those observed along depth gradients: changes in the quantum yield of photosystem II, photosynthetic parameters, and optical properties and pigmentation of the coral tissue. Differences in light availability and photoacclimation driven by surface inclination led to contrasting energetic performance. Horizontally and vertically oriented coral surfaces experienced the largest reductions in photosynthetic usable energy as a result of excessive irradiance and light-limiting conditions, respectively. This pattern is predicted to change with depth or local water optical properties. Our study concludes that colony geometry plays an essential role in shaping the energy balance and determining the light niche of zooxanthellate corals.

## Introduction

The combination of the symbioses with photosynthetic dinoflagellates and colonial integration is thought to be a key factor for the ecological and evolutionary success of corals in tropical coral reef ecosystems [1,2]. The nutritional symbiosis allows this group of sessile benthic animals to efficiently use solar radiation as major source of energy, conferring important

**Funding:** TLL was supported by a scholarship from the Consejo Nacional de Ciencia y Tecnología (CONACYT) from México and by Pennsylvania State University startup funds to RIP. Research funding for SE and RIP was provided by CONACYT-Mexico (Project 129880, Conv-CB-2009). A PASPA fellowship from the DGAPA-UNAM supported the visit of SE to the Biology Department at Pennsylvania State University. The funders had no role in study design, data collection and analysis, decision to publish, or preparation of the manuscript. There was no additional external funding received for this study.

**Competing interests:** The authors have declared that no competing interests exist.

metabolic advantages in the predominantly oligotrophic environments in which they evolved [3,4]. On the other hand, colony integration allows resource translocation from source to sink sites according to metabolic needs [5–8], contributing to their competitive dominance over other benthic invertebrates by strengthening colony-wide fitness and stress resistance [9–11].

Colony integration in corals led to the evolution of a complex array of morphologies under the influence of environmental factors [9]. Changes in colony morphology and surface geometry have important implications on the light interception capacity and the consequent photosynthetic energy acquisition by the whole colony, allowing different coral species to thrive in particular light environments along depth gradients. Until now, the effect of morphology on light interception and energy acquisition has remained largely descriptive and qualitative, albeit with a few notable exceptions including studies primarily focused on branching species from Pacific coral reefs, which have utilized mathematical models based on geometrical laws and proxies for assessing photosynthetic capacity [12–16]. The light interception at a particular colony surface area is mediated by the cosine of the angle between the direction of incident irradiance and that particular surface, a phenomenon based on geometrical relations usually referred as the Lambert's Cosine Law [12,17]. This law states that there is an angle dependent reduction in the amount of light energy intercepted by a surface, following a cosine function. Such physical law may explain why coral species tend to adopt horizontal plate-like morphologies in light-limited environments, such as caves and deep-water, in order to maximize light interception, and branching or foliaceous morphologies in high-light environments to facilitate light attenuation and reduce the coral surface area exposed to intense irradiance and thus the potential of photoinhibition [12,13,15].

Contrasting light climates elicit diverse photoacclimation responses in zooxanthellate corals, as in all primary producers, to minimize the imbalance between the absorbed energy and the energy that can be incorporated through photochemistry or dissipated by photoprotective mechanisms [18]. Readjustments of the photosynthetic apparatus of the zooxanthellae occur over both short- and long-term time scales. In the short-term, the energy imbalance can lead to, *e.g.*, reductions in the energy transfer efficiency to the photosystem II (PSII), which is the protein complex responsible for the release of oxygen and electrons from photosynthetic water oxidation. In the long-term, the energy imbalance can lead to changes in the density of light-harvesting pigments and/or electron-consuming sinks, affecting the photosynthetic electron flow capacity [18,19]. Extensive documentation exists regarding the change in photoacclimatory responses of corals exposed to environmental gradients, primarily focusing on the variation associated with bathymetric differences and species-specific physiological capabilities of symbiotic dinoflagellates [20–24]. Recent research has also highlighted significant variation of photoacclimatory responses associated with seasonal changes in light and temperature [25,26]. Despite these significant advancements in understanding photoacclimation responses in corals, there is a dearth of detailed descriptions regarding changes in the inclination angle of coral surfaces and the associated local light climate. Moreover, the broader implications of these changes for the overall energetic performance of entire coral colonies have not been thoroughly explored.

The amount of light that reaches coral colonies directly affects the photosynthetic activity of *in hospite* zooxanthellae and the costs of repair photodamaged PSII reaction centers, which in turn determines the photosynthetic usable energy that can be translocated to their coral host [27]. Both high- and low-light conditions have the potential to constrain the translocation of usable energy (*i.e.*, carbon-rich photosynthates) and the overall energetic performance of coral holobionts. In areas from coral colonies exposed to high-light intensities, the reduction in the translocation of photosynthetic usable energy can occur due to increased costs of PSII repair in the photosynthetic apparatus of the zooxanthellae. Conversely, in areas exposed to low-light

intensities, such reductions may arise as a consequence of diminished photosynthetic activity and photosynthetically fixed energy [27]. Crucially, a uniform pattern in the rates of PSII photodamage as a function of light exposure has been observed across experiments involving multiple coral species subjected to varying intensity and quality of light [27,28]. The consistency of these trends emphasizes the pivotal role of light intensity in determining the rates of PSII photodamage. Moreover, it implies that the energy costs associated with repairing PSII reaction centers from photodamage as a function of light exposure may be similar among coral and zooxanthellae species. This assumption aligns with the fundamental principle that diurnal recovery of photochemical efficiency is driven by the equilibrium between PSII damage and repair rates [29]. Given the substantial influence of colony morphology on light acquisition, it is plausible that gradients of photosynthetic usable energy exist within colonies, shaped by varying trade-offs between photosynthetic productivity and the costs of repairing PSII from photodamage.

The aim of this study was to investigate the influence of the inclination angle of coral colony surfaces on light exposure, short- and long-term photoacclimatory responses, and the acquisition potential of photosynthetic usable energy. For this purpose, we conducted an *in-situ* experiment with small samples of the coral species *Orbicella faveolata* exposed to different inclination angles. Although this species can form colonies with massive and complex morphologies, the predominant flat morphology at the meso-scale limits the formation of significant surface light gradients, making it an ideal system to study local photoacclimation associated with coral surface inclination. Our findings revealed substantial variations in light availability and photoacclimation responses resulting from changes in coral surface inclination, similar to those commonly observed along depth gradients. Furthermore, we documented here the impact of surface inclination on local productivity, energetic demands and energy fluxes within coral colonies.

## Materials and methods

### Sampling design

Twenty coral fragments of the species *Orbicella faveolata*, each approximately 5x5 cm$^2$ in size, formed the basis for this experimental study. The experimental coral fragments were collected from random colonies at similar depth of 5 m in La Bocana Reef, Puerto Morelos, Mexico (20˚ 52' 33.60" N, 86˚ 51' 2.29" W). The selected donor colonies were spaced at least 5 m apart, potentially originating from different genotypes. The experimental fragments had been used for data collection with non-invasive techniques in previous projects, and had remained undisturbed on a flat, horizontal surface at a constant depth of 5 m for at least one year before conducting the experiment. This ensured that all fragments had a similar photoacclimation state at the beginning of the experiment, as confirmed by comparative analysis of the maximum quantum yield of charge separation of photosystem II (PSII), as well as potentially similar Symbiodiniaceae community composition dominated by *Symbiodinium* spp., *Breviolum* spp., and/or *Cladocopium* spp. [28,30]. Experimental corals were glued with underwater epoxy (Z-Spar A-788 epoxy) to PVC couplers and fixed to a panel at 3 m depth over a seagrass bed to naturally reduce the upwelling irradiance. Coral samples were initially acclimated for two weeks at an intermediate angle of exposure to downwelling irradiance of 45˚, and then evenly distributed across five inclination settings: 0˚ (horizontal), 25˚, 45˚, 65˚ and 90˚ (vertical). The panel was oriented in a north-south direction to ensure a symmetrical diurnal cycle around noon time. The experimental conditions were maintained for one year (April 2014 –April 2015). Temperature at the experimental site was continuously monitored at 5 min intervals throughout the experimental period with HOBO pendant dataloggers (UA-002–64, Onset Computer

Corporation, USA) attached to the PVC panel. Massive coral colonies in their natural habitat at 5 m depth were also used for some analyses. Photoacclimatory responses of both experimental corals and colonies in their natural habitat, were assessed by using non-destructive techniques only.

## Quantum yield of charge separation of PSII

Chlorophyll *a* (Chl *a*) fluorescence was recorded daily in all coral fragments (N = 20, n = 4 per inclination setting) after adjusting the inclination settings, using a submersible pulse amplitude modulated fluorometer (Diving-PAM, Walz, Germany; saturation light pulse width 0.6s of >4500μmol quanta m$^{-2}$ s$^{-1}$). The maximum ($F_v/F_m$) and the effective ($\Delta F/F_m$') quantum yields of PSII were respectively recorded at dusk and at local noon until a steady state of PSII photochemical activity was detected, which required approximately 10 days. The stability of PSII photochemical activity was confirmed by the complete recovery of the maximum quantum yield of charge separation of PSII ($F_v/F_m$), at each inclination setting during a diurnal cycle analysis. This diurnal variation of the quantum yield of PSII was measured from dawn to dusk (06:00–19:00) on a cloudless day at the end of the photoacclimatory period. Measurements of $F_v/F_m$ and $\Delta F/F_m$' were also recorded on coral surfaces along an inclination gradient of three massive *O. faveolata* colonies in their natural habitat at a constant depth of approximately 5 m. *In situ* data were collected along transects facing north laid from top to bottom of the colonies, controlling the inclination angle with a buoyancy device attached to a customized protractor. Only flat coral areas with measurable, consistent inclination angles between 0° and 90° were considered in this analysis to ensure measurement accuracy. All Chl *a* fluorescence measurements were collected after a few seconds of dark adaptation to relax the photochemical quenching (*qP*), achieved by attaching an opaque hose that extended 1 cm beyond the tip of the fiber optics cable. The maximum excitation pressure over PSII ($Q_m$) was calculated as $Q_m$ = 1 - [($\Delta F/F_m$' at noon) / ($F_v/F_m$ at dusk)]. As calculated here, $Q_m$ serves as a proxy for the amount of excessive energy absorbed and dissipated as heat, or non-photochemical quenching (NPQ) at noon [31].

The incident photosynthetic radiation at each inclination setting was also recorded when measuring $\Delta F/F_m$' on the experimental fragments, using the cosine-corrected micro-quantum sensor of the Diving-PAM previously calibrated against a reference quantum sensor (LI- 1400; LI-COR, USA). The sensor was carefully positioned perpendicular to the coral surface at each inclination setting (*i.e.*, pointing in the direction of the normal surface) with the help of a custom-made guiding panel. The relative variation in the direct and diffuse components of irradiance was determined on a single day at noon, following Kirk (17). At each inclination setting, three replicates of the total incident photosynthetic radiation were recorded ($E_t$), each one immediately followed by another measurement of the diffuse irradiance ($E_{df}$) obtained by placing a black panel at a constant distance of 10 cm above the sensor to remove the direct irradiance ($E_{dr}$). Finally, $E_{dr}$ was calculated as $E$t—$E_{df}$.

## Optical properties of coral tissues

The light absorption capacity of intact corals was determined spectrophotometrically following the technique initially developed by Shibata [32] and subsequently modified [33]. The reflectance (*R*) spectra of all experimental corals (N = 20, n = 4 per inclination setting) were measured 10 months after adjusting the inclination settings, using a modular spectrophotometer (Flame-T-UV-VIS, Ocean Optics Inc.). The fraction of incident light absorbed by the coral tissue, absorptance (*A*), was calculated as *A* = 1—*R*, assuming that the light transmitted through the coral skeleton is negligible [33]. The absorptance peak of chlorophyll *a* (Chl *a*) at 675 nm

($A_{675}$) was used as a proxy of relative changes in Chl $a$ content, considering that light absorption at this wavelength has minimal interference from accessory algal and animal pigments [33]. The minimum quantum requirement of photosynthesis ($1/\Phi_{max}$) was calculated based on the amount of light being absorbed and used to drive photosynthesis in the linear increase of the $P$ vs $E$ curve at sub-saturating irradiance (see below). The ratio of absorbed light in the PAR range, or photosynthetic usable radiation (PUR), was calculated by multiplying the light emission spectrum of the lamp used in the $P$ vs $E$ curves characterizations by the *in vivo* absorption spectrum of corals [34,35].

## Photosynthetic parameters

Photosynthetic parameters derived from $P$ vs $E$ (photosynthesis vs irradiance) curves were obtained from 15 experimental corals, 12 months after exposing the samples to the experimental conditions. For the analysis, three coral samples from each inclination setting that maintained predominantly flat surfaces were selected, avoiding the formation of internal light and productivity gradients during measurements. Photosynthesis and respiration rates were measured using a fiber-optic oxygen meter system (FireSting, Pyroscience) with a custom-made acrylic chamber filled with filtered seawater (0.45 μm), which was maintained under constant agitation using magnetic stirrers. Water temperature was maintained at 28˚C using an external circulating water bath (Isotemp, Fisher Scientific, USA). Ten levels of irradiance (0, 37, 65, 97, 147, 232, 361, 479, 677 and 904 μmol quanta $m^{-2}$ $s^{-1}$) were supplied at 10 min intervals with a dimmable LED (Philips, China) and a combination of neutral density filters. To control for photosynthesis and respiration of micro-organisms and biofilm, we covered the non-living portions of corals with black plasticine and repeated the $P$ vs $E$ curve protocol with filtered seawater without corals. Photosynthetic parameters were calculated as in González-Guerrero, Vásquez-Elizondo (35), normalized to coral surface area estimated with the aluminum foil technique [36]. Briefly, the respiration rates ($R$) were calculated by averaging the dark respiration rates obtained before illumination ($R_D$) and after exposing the corals to the maximum level of irradiance ($R_L$). Maximum photosynthetic rates ($P_{max}$) were calculated using the averaged values of photosynthesis obtained above the saturating irradiance ($E_k$). The compensating irradiance ($E_c$) corresponded to the light intensity at which the rate photosynthesis matched the rate of respiration. The slope of the linear increase of photosynthesis at sub-saturating irradiance (or photosynthetic efficiency, $\alpha$), was calculated using least-square regression analyses. This last parameter was also used for the calculation of $1/\Phi_{max}$, correcting the incident irradiance on the coral surface ($E$) for the absorbed light ($A$).

## Local irradiance regimes

Data of sea surface irradiance ($E_0$) recorded by a nearby (~100 m) meteorological station (Estación Meteorológica de la Unidad Académica de Sistemas Arrecifales Puerto Morelos, SAMMO) were used to characterize the temporal variation of solar energy and light-driven processes according to the inclination angle of coral samples (n = 143 days). Total downwelling irradiance at the experimental depth of 3 m ($E_z$) was calculated following the Lambert-Beer law ($E_z = E_0\ e^{-Kd\ z}$) [17], using a vertical attenuation coefficient for downwelling irradiance ($K_d$) of 0.20 $m^{-1}$ [37]. Subsequently, local available irradiance at each inclination setting ($E_\theta$) was estimated using correction factors, which were derived from empirical measurements of local irradiance at noon (1.00 at 0˚, 0.85 at 25˚, 0.58 at 45˚, 0.34 at 65˚, and 0.09 at 90˚). The proportion of daily light integral (DLI) corresponding to sub-compensating (below $E_c$), compensating (between $E_c$ and $E_k$), and saturating light intensities (above $E_k$) were also calculated for each inclination setting using the mean values of the $P$ vs $E$ curve parameters. The daytime

period during which the light intensity exceeded the compensating irradiance ($H_{com}$) and saturating irradiance ($H_{sat}$) were calculated as in Dennison and Alberte [38]. $H_{com}$ was used as descriptor of the daytime period (in hours) during which irradiance was enough for photosynthesis to compensate the respiratory demands of the coral holobiont, while $H_{sat}$ was used as descriptor of the amount of time (in hours) during which irradiance exceeded the maximum photosynthetic capacity of corals.

## Light-driven processes

Based on the availability of light and the mean values of the *P* vs *E* curve parameters at each inclination setting, we estimated the relative change in coral productivity. The photosynthetic productivity ($P_g$) over diurnal cycles was estimated following Jassby and Platt [39] as: $P_g = P_{max} \tanh(\alpha E_\theta / P_{max})$. We assumed that not all algal photosynthates are translocated to their coral hosts, considering the energy expenditure of the zooxanthellae associated with repairing its photosynthetic apparatus from photodamage which is proportional to light availability [27]. We used the same mathematical model outlined in López-Londoño, Gómez-Campo (27) to assess the relative change in photosynthetic usable energy supplied by the zooxanthellae to their coral host (*PUES*) in relation to light exposure based on the inclination angle of coral surfaces. This model assumes that the primary factor contributing to the maintenance costs in the symbiotic zooxanthellae is the light-induced damage of PSII, along with the subsequent synthesis of proteins required for their continual re-assembly. The change in PSII half-time (PSII $t_{1/2}$) as a function of light exposure was calculated with the power function: PSII $t_{1/2} = ME^\Delta$, where *M* represents the maximum theoretical value of PSII $t_{1/2}$, *E* represents the estimated light exposure at each inclination setting, and $\Delta$ denotes the rate of change of PSII $t_{1/2}$ with light availability. The values employed in this analysis for *M* and $\Delta$ were 28.10 and -0.50, respectively, derived from experiments conducted with the Caribbean coral *Porites astreoides* [27]. Despite our study being centered on a different coral species, we chose these values due the notable similar trends observed in the rates of PSII photodamage with light exposure in experiments involving several coral species [27,28]. The consistency of this pattern highlights the paramount influence of light intensity on the rates of photodamage and consequent costs of PSII repair, which appear to be highly consistent among coral species and *in hospite* zooxanthella types. The costs of repair from photodamage ($C_a$) were determined as percentage of the total amount of photosynthetically fixed energy ($P_g$) over a diurnal cycle (12 hours of daylight), through the relation: $C_a = (12 / \text{PSII } t_{1/2}) R$, where *R* represents the relative energy cost of protein turnover for the reassembly of PSII reaction centers (*R* = 15% as in López-Londoño, Gómez-Campo (27)). The *PUES* at each inclination setting was calculated by subtracting the estimated costs of repair from the photosynthetic output of the zooxanthellae (*PUES* = $P_g$—$C_a$) [27].

## Data analysis

Simple linear regression models were used to explore the explanatory power of coral surface inclination angle on the variation of local irradiance and coral physiology and optics. An analysis of covariance was conducted to test for differences in the linear regression describing the association of photosynthetic parameters with coral surface inclination in experimental samples and intact colonies in their natural habitat (interaction of parameters with inclination angle of coral surfaces). Pearson-product correlation analyses were conducted to explore the type and magnitude of the relationship between some variables (respiration vs photosynthetic rates, and maximum quantum yield of PSII vs minimum quantum requirement). Analyses were conducted using R version 4.2.2 [40].

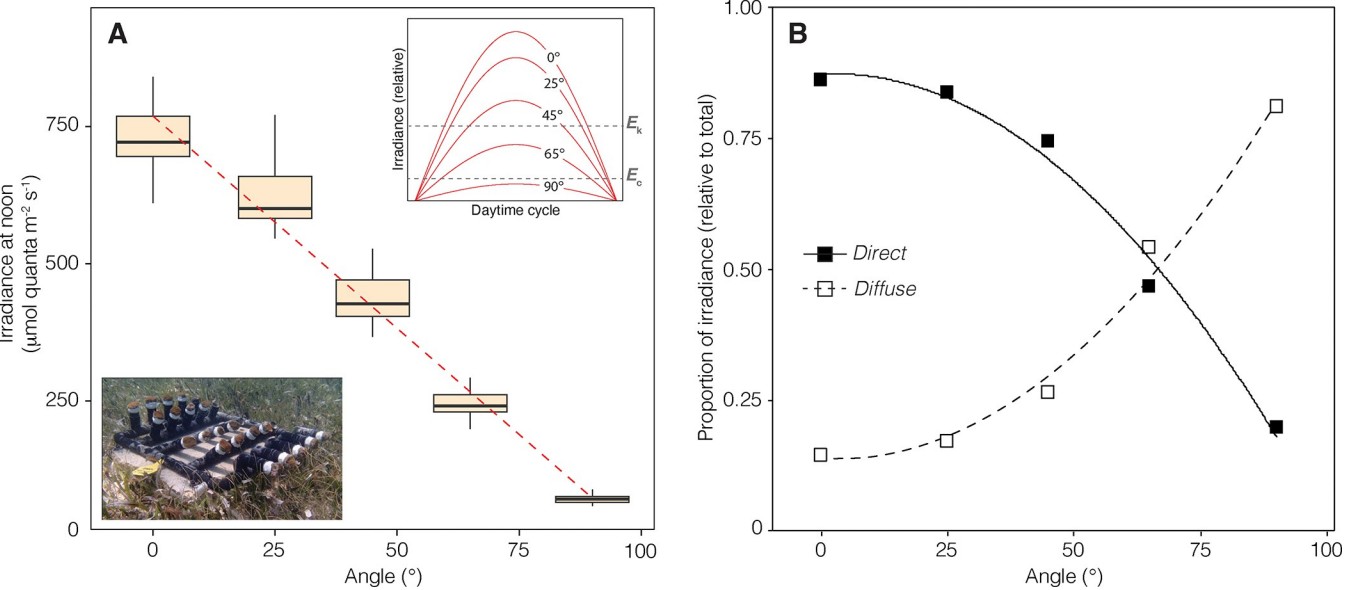

**Fig 1. Variation in the local light climate with the inclination angle of coral surface.** (A) Box plots of the peak irradiance at local noon with a linear regression describing the relationship between irradiance and surface inclination. The top right insert depicts the predicted pattern of diurnal variation of irradiance for each inclination setting, indicating the relative positions of the compensating irradiance ($E_c$) and saturating irradiance ($E_k$). The bottom-left insert shows the PVC structure used in the experiment to adjust the inclination angle of coral samples. (B) Relative variation of the diffuse and direct components of total irradiance. Cosine functions were used to illustrate the relationship between the direct and diffuse components of irradiance with surface inclination.

## Results

Coral samples of *Orbicella faveolata* were exposed to contrasting light climates by adjusting the inclination angle of the coral surface, which spanned from 0˚ (fully horizontal orientation) to 90˚ (fully vertical orientation). Maximum irradiance recorded at local noon available for horizontally oriented corals was 735.67 ± 68.67 (mean ± standard deviation) µmol quanta·m⁻²·s⁻¹, while for vertically oriented corals it was 71.04 ± 8.06 µmol quanta·m⁻²·s⁻¹. In relative units, the available irradiance for corals at 0˚ and 90˚ inclination angles was estimated as 55% and 5% of the incident irradiance at sea surface (%$E_S$), respectively. A linear regression analysis revealed that surface inclination angle explained 95% of the variation in maximum irradiance available for the experimental corals (**Fig 1A**; $R^2$ = 0.95, p < 0.001). The direct and diffuse components of local irradiance exhibited opposite patterns with respect to coral surface inclination (**Fig 1B**). In vertically oriented corals, the diffuse component accounted for most of the available irradiance (81%), while for horizontally oriented corals, it was the direct component that contributed the most to the available irradiance (86%). A strong positive correlation was found between the direct component of local irradiance and the cosine of the angle of coral surface ($r_{(3)}$ = 0.99, p < 0.001). Temperature at the experimental site during the experimental period was 28.72 ± 1.76˚C.

### Quantum yield of charge separation of PSII

After nearly five days of acclimation to the inclination gradient, the maximum excitation pressure over photosystem II (PSII) at noon ($Q_m$) was found to be more than five times higher in horizontally oriented corals (0.56 ± 0.08) than in vertically oriented corals (0.10 ± 0.04) (**Fig 2A**). Absolute values of the maximum ($F_v/F_m$) and effective ($\Delta F/F_m'$) quantum yields of PSII over the diel cycle showed a gradual reduction of the PSII photochemical activity after dawn,

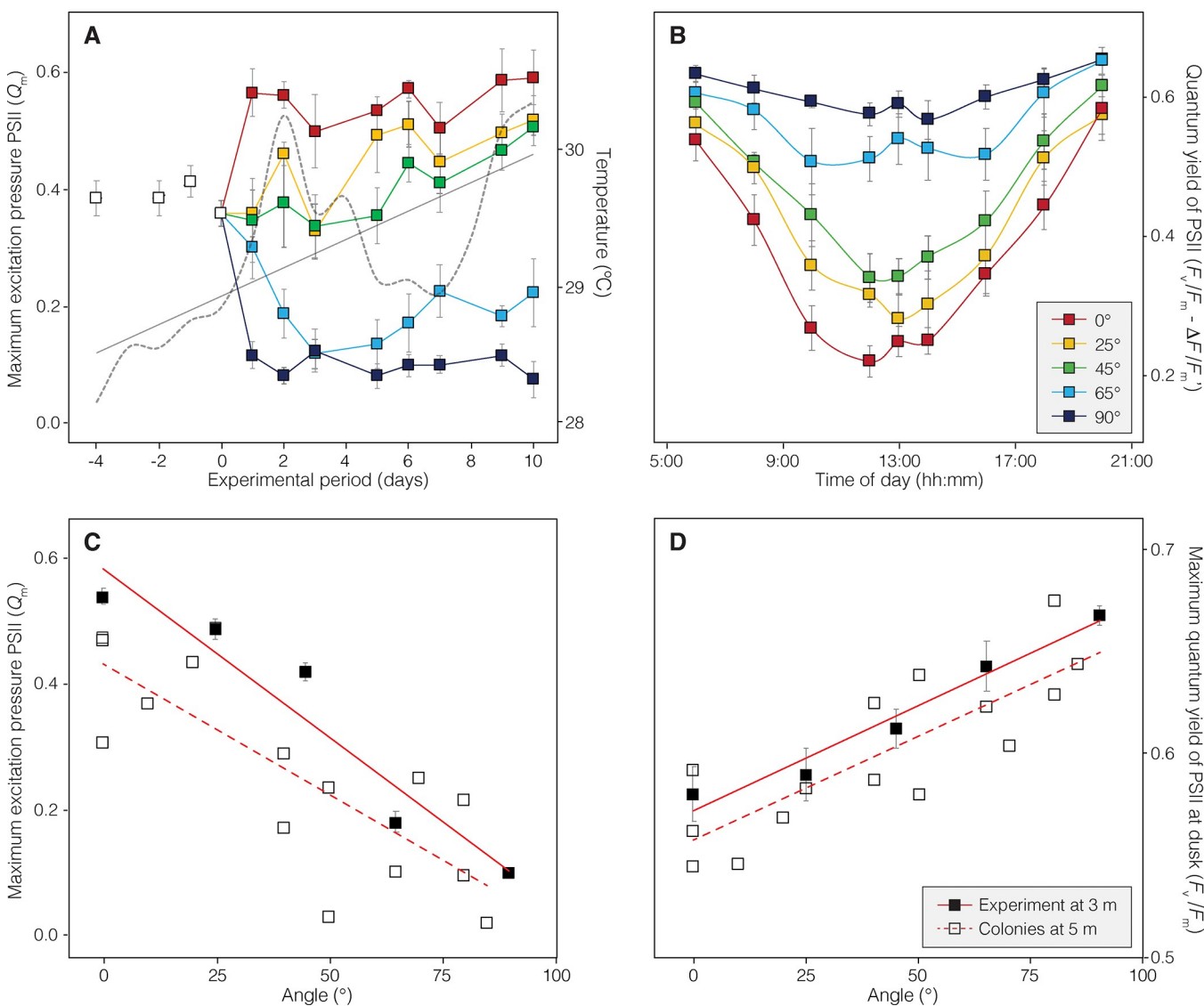

**Fig 2. Variation in chlorophyll *a* fluorescence parameters with coral surface inclination.** (A) Maximum excitation pressure over PSII ($Q_m$) recorded before and after exposure to the experimental conditions until a steady state of PSII photochemical activity was detected. The diurnal variation in temperature (daily mean) is depicted using a gray dashed line, while a linear regression of the temperature variation is presented as a solid line.(B) Diurnal oscillation of the maximum ($F_v/F_m$) and effective ($\Delta F/F_m'$) quantum yield of the PSII. The linear associations of $Q_m$ and $F_v/F_m$ with coral surface inclination are shown in plots (C) and (D) respectively, for the experimental samples at 3 m depth (solid squares and continuous lines) and field measurements on coral colonies in their natural habitat at 5 m depth (empty squares and discontinuous lines). Error bars in all graphs are SE.

reaching minimum $\Delta F/F_m'$ values at midday and then gradually recovering at dusk until reaching former dawn $F_v/F_m$ levels in all inclination settings. However, important differences were observed in the maximum reductions of $\Delta F/F_m'$ at noon relative to maximum $F_v/F_m$ values at dusk/dawn along the inclination gradient, being much more pronounced in horizontally oriented corals (average reduction of 62%) compared to vertically oriented corals (average reduction of 13%) (**Fig 2B**, see **S1 Table** for all parameters estimates). During the initial acclimation phase to the experimental conditions, corals in certain inclination settings, primarily those at intermediate inclination angles, did not reach a clearly defined steady-state in the photochemical activity of PSII. This phenomenon coincided with a gradual temperature increase observed during this initial photoacclimation phase (Fig 2A).

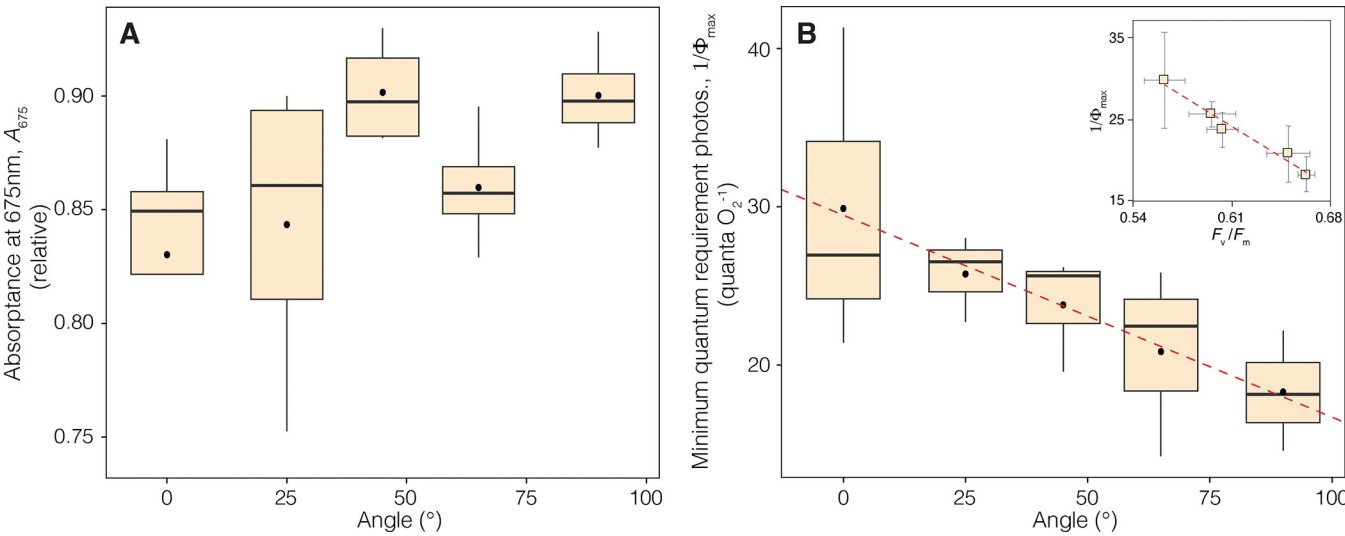

**Fig 3. Variation in optical properties of the coral tissues and associated photosynthetic descriptors with coral surface inclination.** (A) Boxplots describing the variation in coral absorptance at 675 nm ($A_{675}$), and (B) minimum quantum requirements for oxygen evolution ($1/\Phi_{max}$) with coral surface inclination. The insert in (B) depict the linear relationship between $1/\Phi_{max}$ and maximum quantum yield of PSII ($F_v/F_m$), with error bars as SE.

Significant linear associations between all photosynthetic descriptors derived from chlorophyll $a$ (Chl $a$) fluorescence measurements and coral surface inclination angle were also found, both for colonies under natural environmental conditions and for the experimental samples. Linear regression analyses indicated that the inclination angle of coral surface explained 78% of the $Q_m$ variation in experimental corals at 3 m depth and 63% of the within-colony variation in natural environments at 5 m depth ($R^2 = 0.78$, $p < 0.001$ and $R^2 = 0.63$, $p < 0.001$, respectively) (**Fig 2C**). Coral surface inclination also explained 68% of the within-colony variation of $F_v/F_m$ in natural environments ($R^2 = 0.68$, $p < 0.001$) and 36% of the $F_v/F_m$ variation in experimental samples ($R^2 = 0.36$, $p < 0.001$) (**Fig 2D**). The covariance analysis (condition by angle interaction) indicated that the pattern of change in both $Q_m$ and $F_v/F_m$ as a function of coral surface inclination was similar in natural environments and experimental conditions ($F_{(1,91)} = 1.95$, $p = 0.17$ and $F_{(1,91)} = 0.001$, $p = 0.97$, respectively).

## Optical properties of coral tissues

Variation in coral absorptance at 675 nm ($A_{675}$), which corresponds to the peak absorption wavelength of chlorophyll $a$ (Chl $a$), was employed as a non-destructive proxy to assess changes in Chl $a$ content in the experimental corals. This proxy indicated that coral pigmentation also varied as a function of the inclination angle of coral surfaces. Linear regression analysis revealed that the inclination angle accounted for only 21% of the variation in $A_{675}$, with a marginal level of significance ($R^2 = 0.21$, $p = 0.042$). The greatest reductions in $A_{675}$ were observed in horizontal and slightly inclined coral samples ($0.83 \pm 0.06$ at 0° and $0.84 \pm 0.07$ at 25°), which is consistent with a decrease in pigment content. In contrast, the highest values were observed in corals at intermediate and greater inclination angles ($0.90 \pm 0.02$ at 45° and 90°), suggesting maximum content of photosynthetic pigments in these samples (**Fig 3A**).

The inclination angle of the coral surface explained 40% ($R^2 = 0.40$, $p = 0.011$) of the variation in the minimum quantum requirement of photosynthesis ($1/\Phi_{max}$). On average, the values of this parameter were 63% lower in vertically oriented corals exposed to the lowest irradiance ($18.31 \pm 3.79$ quanta $O_2^{-1}$) compared to horizontally oriented corals exposed to the

highest levels of direct sunlight (29.90 ± 10.27 quanta $O_2^{-1}$) (**Fig 3B**). Furthermore, a significant negative correlation was observed between the averaged values of $1/\Phi_{max}$ and $F_v/F_m$ at each inclination setting ($r_{(3)}$ = -0.98, p = 0.002) (**Fig 3B**).

## Photosynthetic parameters

Photosynthetic parameters derived from the *P* vs *E* curves were also variable in experimental corals, and a significant portion of this variation was found to be explained by the surface inclination angle. Linear regression analyses indicated that the inclination angle explained 48%, 32%, and 38% of the respective variation in the slope of the linear increase in photosynthesis at sub-saturating irradiance (or photosynthetic efficiency, $\alpha$) ($R^2$ = 0.48, p < 0.01), compensating irradiance ($E_c$) ($R^2$ = 0.32, p = 0.03) and saturating irradiance ($E_k$) ($R^2$ = 0.38, p = 0.01) (**Fig 4A–4C**). On average, $\alpha$, $E_c$ and $E_k$ were respectively 42% lower, and 96% and 70% higher on

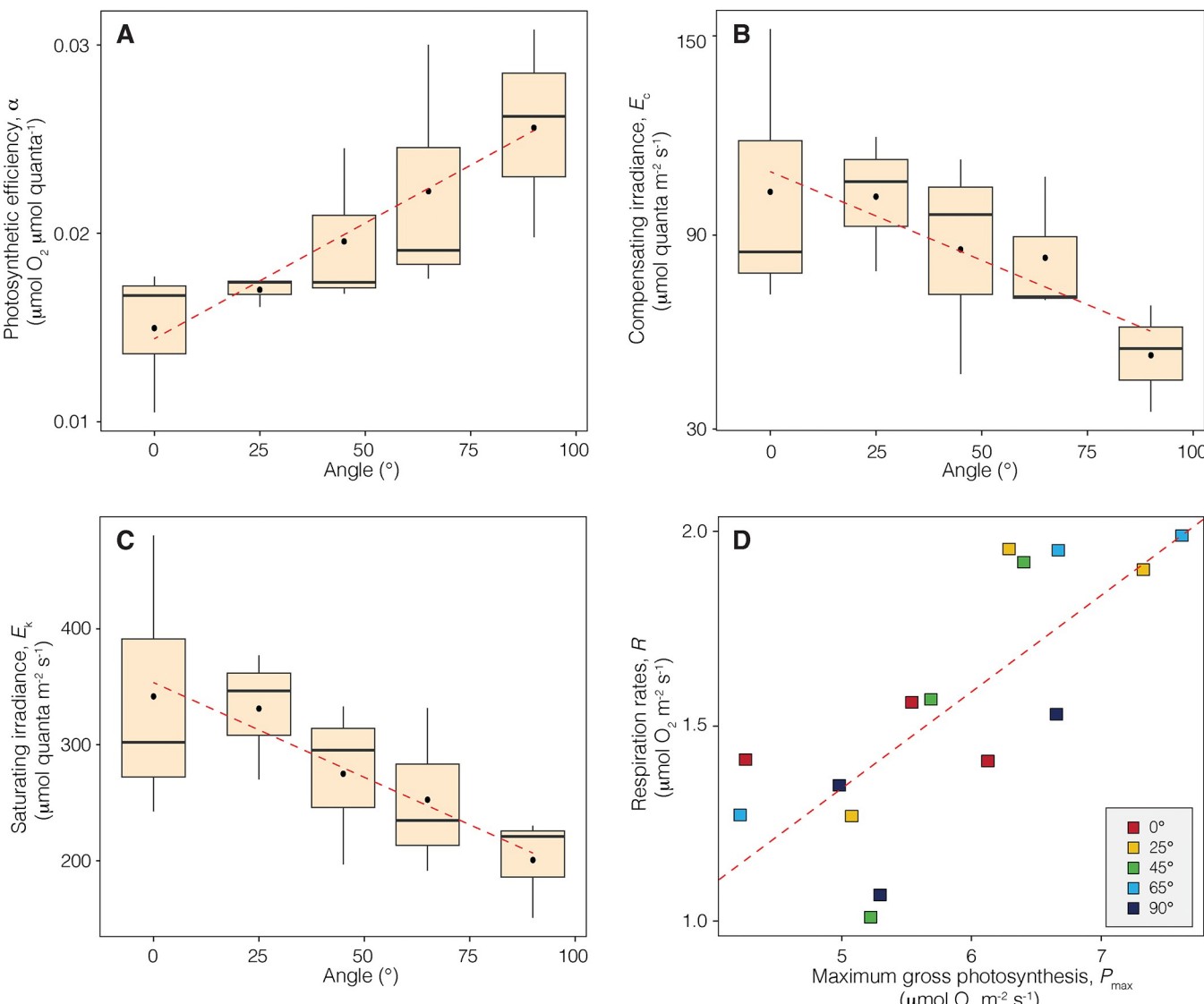

**Fig 4. Variation in photosynthetic parameters with the inclination angle of coral surface.** (A-C) Boxplots describing the variation of three photosynthetic parameters ($\alpha$, $E_c$ and $E_k$), along with linear regressions showing the association between each parameter and coral surface inclination. (D) Linear association between $P_{max}$ and $R$ of all coral samples analyzed in this study.

horizontally oriented corals compared to vertically oriented ones. No significant association was observed between the variation in maximum photosynthesis ($P_{max}$) and respiration rates ($R$) with coral surface inclination ($R^2 = 0.01$, p = 0.75 and $R^2 = 0.01$, p = 0.67, respectively). However, a strong positive correlation (Pearson-product) was found between $P_{max}$ and $R$, regardless of coral surface inclination ($r_{(13)} = 0.76$, p < 0.001) (**Fig 4D**). Similarly, a strong positive correlation was detected between the dark respiration rates obtained before illumination (pre-illumination respiration, $R_D$) and after exposing the corals to the maximum level of irradiance (post-illumination respiration, $R_L$) ($r_{(13)} = 0.61$, p = 0.01).

## Local irradiance regimes

During 143 days of irradiance data and assuming a constant photoacclimatory condition in the experimental samples, the daily light integral (DLI) received by corals in horizontal position was estimated to be $17.07 \pm 4.36$ mol quanta m$^{-2}$ d$^{-1}$, while for corals in vertical position, the DLI was reduced to $1.62 \pm 0.41$ mol quanta m$^{-2}$ d$^{-1}$. The different components of the DLI which corresponded to sub-compensating (below $E_c$), compensating (between $E_c$ and $E_k$) and saturating irradiance (above $E_k$) exhibited high variability along the inclination gradient (**Fig 5A and 5B**). Nearly 1/3 of the DLI experienced by horizontally oriented corals at 0° exceeded $E_k$, corresponding to solar energy absorbed in excess by the photosynthetic apparatus of the zooxanthellae. The amount of excess solar energy absorbed gradually decreased with increasing coral surface inclination until reaching negligible values in corals at an inclination angle of 65°, where it represented less than 1% of the DLI.

In contrast, most of the DLI available for vertically oriented corals at 90° (90%), corresponded to sub-compensating levels of irradiance (*i.e.*, below $E_c$) (**Fig 5B**). Accordingly, the estimated periods of daytime when light intensity exceeded the compensating irradiance ($H_{com}$) and saturating irradiance ($H_{sat}$), were highly variable along the inclination gradient. On average, the $H_{com}$ period for corals in horizontal position accounted for approximately 70% of the daytime ($9.85 \pm 1.60$ h), while for vertically oriented corals it was reduced to 30% ($4.16 \pm 1.78$ h). Notably, on some days, corals at inclination angles of 65° and 90° experienced

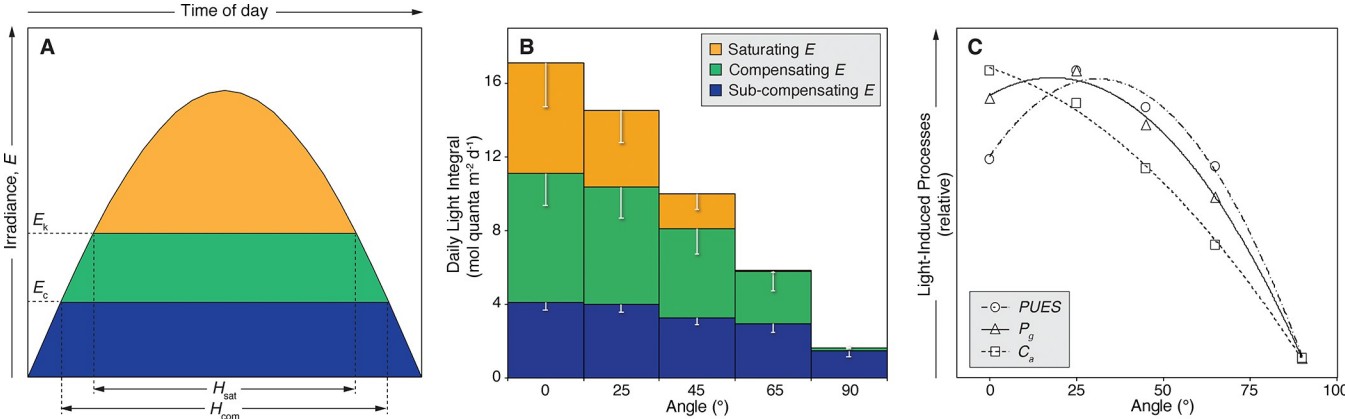

**Fig 5. Local irradiance regimes and light-driven processes according to coral surface inclination.** (A) Description of a diurnal cycle of solar irradiance ($E$), indicating the relative positions of the compensating irradiance ($E_c$) and saturating irradiance ($E_k$), and the daytime periods of light intensities exceeding $E_c$ ($H_{com}$) and $E_k$ ($H_{sat}$) [modified from [38]]. (B) Estimated daily light integral (DLI) at each inclination setting, indicating the relative proportions corresponding to sub-compensating, compensating, and saturating irradiance. Error bars describe standard deviations. (C) Estimated relative changes in light-induced processes affecting the energy balance of coral holobionts. *PUES*: photosynthetic usable energy supply; $P_g$: gross productivity of the zooxanthellae; $C_a$: metabolic costs of maintenance associated with zooxanthellae photosynthesis. Polynomial regressions were used to depict the relationships of these parameters with surface inclination angle.

light-limiting conditions throughout the entire day, where irradiance levels never reached $E_c$. The $H_{sat}$ period for horizontally oriented corals comprised 48% of the daytime (6.70 ± 2.18 h), whereas for corals at 65˚ it was reduced to 3% (0.44 ± 0.69 h). The DLI analysis indicated that vertically oriented corals never reached maximum photosynthetic capacity.

## Light-driven processes

The estimated photosynthetic production ($P_g$) exhibited the highest values in corals at inclination angles of 25˚ (0.17 ± 0.03 mol $O_2$ $m^{-2}$ $d^{-1}$) and 0˚ (0.15 ± 0.03 mol $O_2$ $m^{-2}$ $d^{-1}$), while the lowest values were observed in corals at an inclination angle of 90˚ (0.04 ± 0.001 mol $O_2$ $m^{-2}$ $d^{-1}$) (**Fig 5C**). The estimated energy costs of repair from photodamage for the symbiotic algae ($C_a$), expected to be proportional to light availability [27], ranged between 49% and 15% of the photosynthetically fixed energy at both ends of the inclination gradient. Consequently, the photosynthetic usable energy supplied (*PUES*) by the symbiotic algae to their coral host followed a unimodal pattern across the inclination gradient, being higher in coral surfaces at 25˚ and 45˚ inclination angles. The *PUES* calculated for fully vertically oriented corals exhibited the lowest values along the experimental inclination gradient (**Fig 5C**).

## Discussion

Colony surface geometry generates, within a single depth, a diverse array of local light climates that led to significant variation in the photoacclimatory responses of the coral holobiont and in significant differences in the photosynthetic usable energy supplied (*PUES*) by the symbiotic algae to their coral host. Increasing the inclination angle of coral surface results in a proportional reduction of light availability, influenced by relative changes in the diffuse and direct components of total irradiance. The strong positive correlation between the direct component of irradiance and the cosine of the angle of coral surface confirms that coral surfaces behave as a cosine-corrected light collectors at this scale [33], in accordance with Lambertian cosine law [17]. While changes in light availability are commonly associated with depth gradients and the optical properties of the water column, the impact of colony surface geometry on light availability for the holobiont is often overlooked, albeit with a few notable exceptions in studies predominantly conducted on branching species from Pacific coral reefs [12–16]. The extremes within the light intensity gradient observed in this study as a result of changes in colony surface inclination were comparable to the estimated changes in downwelling irradiance across depths separated by approximately 40 m in clear waters [41] or between open habitats and groove formations [42]. Consistent with previous studies [12,13,15], the results of this research provide compelling evidence that the light gradients taking place along broad depth ranges can also manifest within a single colony, and that this variation is modulated by colony morphology. The variability in these local light climates elicits comparable photoacclimation responses to optimize the balance between absorbed light energy, photosynthetic capacity, photodamage and repair processes, and the translocation of photosynthetic energy by the symbiotic algae to their coral host.

   After nearly 10 days of exposure to the experimental conditions, all corals exhibited a complete recovery of the maximum quantum yield of PSII ($F_v/F_m$) at the end of the day relative to former dawn levels at each inclination setting, which supports the notion that the PSII had already undergone stable photoacclimation to the local light climate associated with each inclination setting. The light gradient experimentally induced by changing coral surface inclination resulted in contrasting levels of PSII excitation pressure, which is one of the first chloroplastic redox signal to trigger photoacclimatory changes in primary producers [18,43]. Samples with higher inclination angles exposed to lower light intensities (at 65˚ and 90˚) exhibited minimal

diurnal oscillations in PSII quantum yields and, consequently, minimal values of maximum excitation pressure over PSII ($Q_m$). This pattern suggests that even during periods of maximum irradiance most PSII reaction centers remain open, indicating light-limited photosynthesis and low energetic contribution by the symbiotic zooxanthellae to their coral host [31]. In contrast, coral samples with lower inclination angles (at 0˚ and 25˚) exposed to higher levels of direct sunlight exhibited substantial diurnal oscillations in PSII quantum yields, reflected in higher values of $Q_m$. This indicates that during peak-irradiance periods, a significant portion of PSII reaction centers becomes closed due to the high rate of energy absorbed (*i.e.*, light dose), requiring maximum induction of photoprotective mechanisms to dissipate as heat the excess of absorbed energy [31,44]. The absence of a clearly defined steady-state in the photochemical activity of PSII in certain experimental conditions, especially at intermediate inclination angles, may be attributed to the influence of gradual increases in both light exposure and temperature during the spring season, when $Q_m$ measurements were recorded [26].

Accordingly, variation in $F_v/F_m$ along the inclination gradient reflects a variable proportion of photoinactive PSII reaction centers that are not involved in photochemistry but are better suited to dissipate excessive excitation energy [29,45]. The accumulation of these photoinactive PSII reaction centers is associated with a diurnal balance between the rates of PSII damage and repair under specific light climates. For instance, when the rates of PSII damage exceeds those of repair, there is accumulation of photodamage which is reflected in reductions of $F_v/F_m$. The observed linear association between $F_v/F_m$ and surface inclination angle suggests that the subpopulation of photoinactive PSII accumulated in the algal photosynthetic apparatus may be inversely proportional to the inclination angle of coral surface, resulting in maximum photoprotection on horizontal surfaces and maximal photochemical efficiency on vertical surfaces. The consistency between the patterns obtained in experimental samples and in coral colonies within their native environments suggests that the short-term photoacclimation responses related to the photochemical efficiency of PSII induced by changes in coral surface inclination, likely reflect natural processes occurring in coral colonies with complex geometries [29,43]. It is worthy to remark that these photoacclimation responses are attributed exclusively to variations in total light intensity, rather than changes in light quality (*e.g.*, UV wavelengths) [45] or other environmental conditions (*e.g.*, temperature) [46], due to the consistent experimental depth maintained in our study. This underscores the paramount influence of light intensity on photoacclimation responses linked to accumulation of photodamage, in addition to well-known effect of UV exposure on photosynthetic processes, which can be more pronounced at shallow depths [45,47].

The induction of long-term photoacclimatory responses in coral samples was supported by the observed variation in their *in vivo* optical properties (absorptance/pigmentation) and photosynthetic potential (*P* vs *E* curve parameters). Although the relationship between light absorption capacity ($A_{675}$) and coral surface inclination angle was weak, it still indicated a reduction in pigment content in coral samples with lower inclination angles that were exposed to higher levels of direct sunlight, and that exhibited higher PSII excitation pressure. Such inferred changes in pigmentation along the inclination gradient are consistent with the induction of a photoacclimatory response to reduce excitation pressure on PSII in high light environments and to maximize the light harvesting capacity in low light conditions. Coral samples exposed to higher inclinations (above 45˚) displayed similar $A_{675}$ values close to 0.9, which is the maximum light absorption capacity (*i.e.*, absorptance) at 675 nm documented for *O. faveolata* [48]. This suggests that the light absorption capacity of coral holobionts exposed to higher inclinations was already close to its maximum and minimally influenced by further changes in pigment content. This restricted variation of $A_{675}$ at its maximum value may explain the limited capacity of this proxy for explaining the pigment content variation in the experimental

corals. All these findings highlight the importance of absorptance variation associated with changes in pigmentation as a crucial photoacclimatory response to modulate light dose, not only for coral colonies located at different depths [20,21,23] but also for coral surfaces within a single colony exhibiting contrasting inclination.

Comparative analyses of the *P* vs *E* curve parameters revealed clear linear associations of the inclination angle of the coral surface with the photosynthetic efficiency ($\alpha$), compensating irradiance ($E_c$), and saturating irradiance ($E_k$). The pattern of variation in these parameters is similar to that observed in corals growing under contrasting light climates along depth gradients [20,21,23], being directly associated with the observed changes in absorptance/pigmentation and quantum yield efficiency of PSII. Several outcomes result from the link between these parameters and coral surface inclination. Firstly, the enhanced photosynthetic efficiency ($\alpha$) in coral surfaces with greater inclination angles leads to more rapid increases in photosynthetic rates with increasing light availability. Secondly, the lowered compensating irradiance ($E_c$) in coral surfaces with greater inclination angles and limited light availability enhances their capacity to achieve positive energy balances over extended periods of the daytime. Lastly, the elevated saturating irradiance ($E_k$) in horizontal or slightly inclined coral surfaces exposed to higher levels of direct sunlight leads to reductions in the proportion of daytime when irradiance exceeds the maximum photosynthetic capacity, minimizing the accumulation of excessive energy absorbed that cannot be used in photosynthesis and must be dissipated through photoprotective mechanisms. Both $E_c$ and $E_k$ are very sensitive to changes in the photosynthetic efficiency ($E_c = R/\alpha$ and $E_k = P_{max}/\alpha$). Therefore, variations in these photosynthetic parameters along the inclination gradient primarily are likely attributed to changes in the absorption cross-section of the coral surface and/or the number of PSII reaction centers in the photosynthetic apparatus of the zooxanthellae, which directly influence light utilization efficiency [18,49]. It is worth noting that these optical and photosynthetic properties should be attributed to the coral holobiont (the coral animal + symbiotic algae + coral skeleton) rather than exclusively to Symbiodiniaceae.

The strong correlation between $P_{max}$ and $R$, independent of the inclination angle of the coral surface, suggests that both parameters responded similarly to the local environmental conditions, although did not vary linearly along the inclination gradient examined in this study. The constraints of a limited sample size at each inclination setting and our commitment to non-destructive techniques in our experiment, prevented a more in-depth analysis of the observed correlation between $P_{max}$ and $R_d$ along the experimentally induced light gradient. It is also worth highlighting that potential variations in symbiont cell density, cell size, algal genotype and/or coral tissue thickness (*i.e.*, biomass), among other structural traits not accounted for in this study, can be important underlying factors for the strong correlation between both $P_{max}$ and $R$, and between the minimum (before illumination, $R_D$) and maximum (after exposing corals to the maximum level of irradiance, $R_L$) rates of respiration. These strong correlations may indicate that some of these unaccounted coral traits play a crucial role in determining the maximum metabolic activity and photosynthetic potential among samples, as has been observed in corals acclimated to contrasting coral reef habitats [21,42,50] and seasons [25,26].

The analysis of the interplay between photosynthetic parameters and optical properties of coral holobionts offers a unique opportunity to assess an essential descriptor of photosynthetic performance: the minimum quantum requirement of photosynthesis ($1/\Phi_{max}$), which measures the minimum number of absorbed photons required to evolve one molecule of $O_2$ [34,35]. Analyses of this parameter revealed a clear pattern of change within the experimental corals, showing an inverse relationship with the inclination angle of coral surfaces (*i.e.*, as the inclination angle increased, the values of $1/\Phi_{max}$ decreased), mirroring the variation observed

in several coral species in response to changes in depth [21,51]. This trend indicates that coral surfaces with high inclination angles and reduced light availability become more efficient at using the absorbed light energy for photosynthesis than coral surfaces with low inclination angles and maximum light exposure. Furthermore, the correlation between $1/\Phi_{max}$ and $F_v/F_m$ reinforces the fundamental role of the accumulation of photodamaged PSII reaction centers associated with light exposure in influencing the photosynthetic quantum efficiency of corals.

Analyses of the diurnal variation in irradiance along the inclination gradient revealed contrasting effects on the photosynthetic performance of coral holobionts. Coral surfaces with low inclination angle, directly exposed to downwelling irradiance, received sufficient light to reach maximum photosynthesis during nearly half of the daytime. The estimated photosynthetic productivity of these coral surfaces remained relatively constant regardless of slight changes in the inclination angle and light exposure. This plateau in photosynthetic productivity occurs after reaching maximum photosynthetic capacity determined by the hyperbolic response of photosynthesis to light availability, where further increases in productivity are not observed once photosynthesis is light-saturated [52]. As a result, a considerable proportion of the daily light dose absorbed by horizontal and nearly horizontal coral surfaces cannot be utilized for photosynthetic energy conversion but, instead, represents excess excitation energy that can be harmful for the photosynthetic apparatus of the zooxanthellae [27,43]. Conversely, coral surfaces with higher inclination angles and reduced light exposure may only receive enough light to exceed the compensating irradiance during brief periods of the day and may never reach their maximum photosynthetic capacity. Moreover, our results suggest that coral surfaces with greater inclination angles can sustain positive energy balances only over short time periods when the photosynthetically fixed carbon exceeds the holobiont respiratory requirements. It is important to note that our analysis of the photosynthetic performance of corals as a function of the inclination angle of coral surfaces is based on photosynthetic parameters normalized to surface area. However, considering potential variations in unaccounted coral traits, different normalizations beyond surface area (*e.g.*, to host protein, symbiont cell density or size, chlorophyll content) can yield distinct patterns of coral adjustments with significant differences for the beneficial effects on host productivity, depending on the bio-optical and bio-physical characteristics of both the coral host and its algal symbionts [24,26,48].

The non-linearity between light availability, photosynthetic production, and costs of repair from photodamage can result in contrasting levels of photosynthetic usable energy supplied by the zooxanthellae to their host (*PUES*), depending on the geometry of coral colonies. While photosynthetic production follows a hyperbolic response to light intensity, the amounts of absorbed light energy and the costs of repair from photodamage increase proportionally with light availability [27,52]. This energetic imbalance explains the predicted decrease in *PUES* in both vertically and horizontally oriented coral surfaces, similar to changes along depth gradients [27]. The predicted pattern of change in *PUES* as a function of the inclination angle exhibits a unimodal shape, mediated by two contrasting processes: 1) in horizontally oriented surfaces, exposure to intense irradiance amplifies the metabolic costs of maintenance in the zooxanthellae, limiting the amount of energy that can be translocated to their host; and 2) as the inclination angle increases, the reductions in light availability and photosynthetic activity lead to decreases in the amount of energy fixed in photosynthesis. In the shallow experimental site of our study, corals can achieve maximum energetic output from the zooxanthellae and energetic performance at intermediate surface inclination angles. However, with increasing depth, the optimal inclination angles for energetic performance tend to approach the horizontal orientation due to the reduction in the excess of solar energy absorbed relative to the maximum photosynthetic capacity. Heterotrophic feeding plays a fundamental role in corals' nutrition supplying a significant portion of essential elements that cannot be acquired through

autotrophy [3,53]. Some coral species are even able to increase their metabolic reliance on heterotrophy to compensate for reduced photosynthesis under low-light conditions, such as deep areas, caves, or turbid environments [54,55]. Therefore, although our study could not incorporate this crucial aspect of the coral-algae symbiosis, its potential effect on the variation of energy budgets and fluxes within coral colonies cannot be discharged.

In our study, we used small coral samples detached from the original colony. In this sense, it is important to note that under natural conditions, coral colonies with complex morphologies are physiologically integrated, enabling the translocation of energetic products from source to sink sites, such as growing tips and regenerating areas [5–8]. This translocation allows maintaining the highest calcification rates in tips of branching corals, which typically have low zooxanthellae densities and rates of net primary productivity [5,8]. In massive corals, the growth of colony margins and tissue regeneration is similarly promoted by the translocation of photosynthetic products from the most productive areas of the colony [6,7]. Our study was focused on the photoacclimation variability of individual samples and did not directly assess within-colony translocation. However, the evidence from previous studies indicates that the intracolonial translocation of photosynthetically derived energy products may be a common feature among colonial corals, occurring from highly productive source sites to regions with higher metabolic demands. Our results align with the evidence that coral colonies with complex geometries exhibit great heterogeneity in terms of local productivity, energetic demands and energy fluxes, which supports the importance of colony integration and the translocation of energetic products in enhancing the overall performance of the entire coral colony [9–11].

In shallow, high-light environments, horizontal coral surfaces, which are typically located in the central areas of colonies, may not coincide with the most productive regions of the colony due to the increased energy costs of repair from photodamage in the zooxanthellae, which ultimately affects the amount of photosynthetic usable energy supplied to their coral host (*PUES*) [27]. The greater extension and calcification commonly observed in these areas [56] potentially result from photosynthetic energetic products imported from other more productive source sites within the colony. On the other hand, coral surfaces with a large inclination angle and reduced productivity may also act as energy sinks within the colony. Survival of these surfaces would require energetic subsidies from more productive areas of the colony due to their extremely low levels of *PUES* locally available. As depth increases, these low-productivity areas with greater inclination angles are predicted to experience negative photosynthetic energy balances constantly, becoming increasingly less productive and more costly to maintain for the colony. The reduced productivity of colony surfaces with greater inclination angles, along with their partial dependence on other parts of the colony for energy, may partially explain their higher vulnerability to reduced light penetration associated with increased water turbidity [57] (**Fig 6A**), disease outbreaks such as the recent stony coral tissue loss disease [58], as well as environmental perturbations, including heat stress [59,60] (**Fig 6B**).

Overall, the combination of water optical properties [16,57,61], substrate type and landscape architecture [42,62], determines the balance between direct and diffuse components in the local light field at specific depths. The vertical attenuation coefficient for downwelling irradiance ($K_d$), a proxy of the water optical properties, is influenced by the concentration and composition of dissolved and particulate materials, which can absorb and scatter light as it travels through the water. As light penetrates deeper into the water column, there is an increased likelihood of scattering, resulting in an increased proportion of the diffuse component (*i.e.*, the light coming from aside) with depth [17]. This effect may favor light interception and energy acquisition in non-horizontal surfaces of coral colonies located at intermediate depths, where light is still not limited [16,41]. Additionally, different substrates have varying

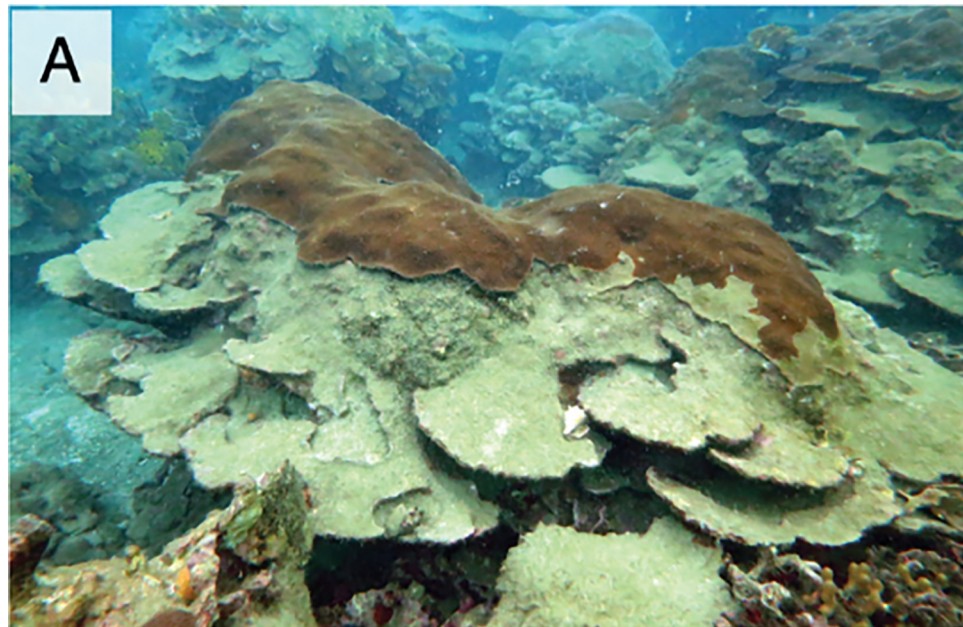

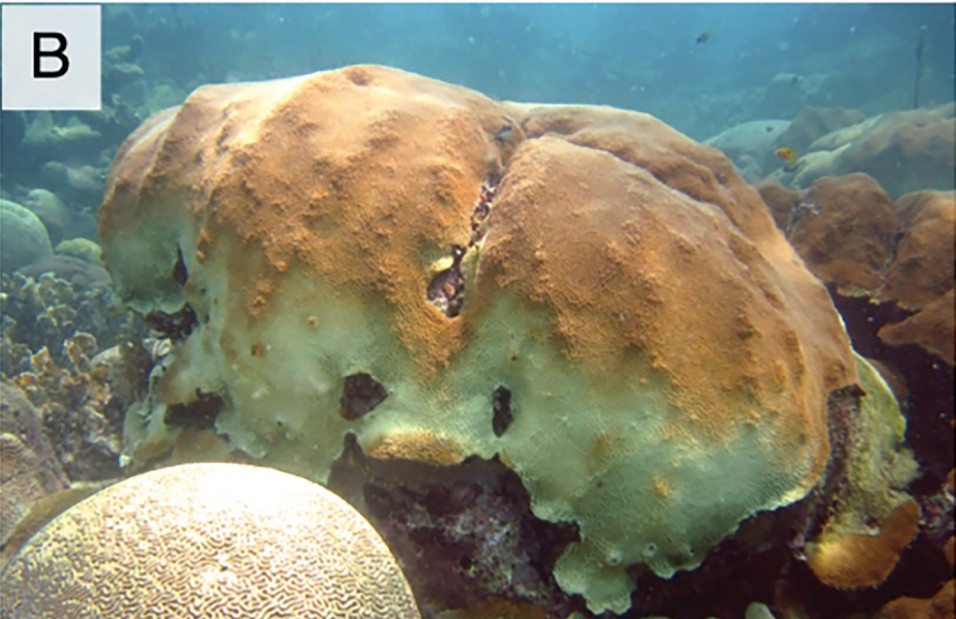

**Fig 6. Coral colonies of *O. faveolata* affected by environmental perturbations.** (A) Partial mortality observed in a coral colony in the Varadero Reef, an area experiencing elevated anthropogenic turbidity [57]. (B) Bleached and non-bleached regions within a colony during the widespread bleaching event that occurred in the Caribbean in 2010. It is noteworthy that in both conditions, the most affected areas are the low-productivity colony surfaces with the larger inclination angles.

reflectance and absorption properties, further influencing the local light regime experienced by benthic photosynthetic organisms, and thus their primary productivity [42,62]. We acknowledge that all these factors along with others, can contribute to the complexity of underwater light fields [63] as well as the diversity of coral responses under apparently similar environmental stressors. However, the framework of this study specifically focuses on investigating the effect of variations in light availability associated with surface inclination on the

photoacclimatory response of coral holobionts, minimizing the potential influence of other biotic and abiotic conditions that can also modulate colony light availability.

In conclusion, our study provides solid support that colony geometry determines light availability, holobiont physiology and photosynthetic energy acquisition in zooxanthellate corals. Previous research has shown that variations in photoacclimation responses of coral holobionts can be linked to changes in the composition of symbiotic zooxanthellae along environmental gradients [22,31,59,64]. Most corals with broad bathymetric distributions, including *O. faveolata*, exhibit changes in the symbiont community composition which presumably allow them to photoacclimate to particular local light environments, both across depths and within colonies [30,59]. Thus, part of the observed variation in holobiont physiological parameters within and between inclination settings, can potentially be attributed to changes in Symbiodiniaceae community composition or other factors related to particular metabolic conditions of experimental samples. However, a significant portion of this variation can be solely attributed to the inclination angle of the coral surface and the associated light climate experimentally induced in this study. This indicates that light energy availability remains a major driver of variation in coral holobiont physiology. Our study provides clear evidence that contrasting photoacclimatory responses, light climates and energy balances can occur within a single colony at a constant depth. Therefore, studying the entire coral colony and its metabolic gradients should be a research priority to enhance our understanding of colonial responses to environmental cues and the role of colony morphology and energetic economy on the vertical distribution of species and niche diversification in coral reefs.

## Supporting information

**S1 Table. Parameters associated with the local light climate and the photoacclimatory responses of corals exposed to the five inclination settings (0˚, 25˚, 45˚, 65˚, and 90˚).** Values correspond to the mean ± S.D. (sample size).
(XLSX)

## Acknowledgments

We thank Claudia T. Galindo-Martínez, Kelly Gómez-Campo, Luis A. González-Guerrero, Ricardo I. Cruz-Cano, Nancy Escandón-Flores, Nadine Schubert and Darren Brown for their valuable assistance during field activities and laboratory analysis. We also thank the Servicio Académico de Monitoreo Meteorológico y Oceanográfico (SAMMO), from the Universidad Nacional Autónoma de México, for providing the data of sea surface irradiance. The National Commission on Aquaculture and Fisheries (CONAPESCA) for research permit (DGOPA 08606.251011.3021). We also thank the DGAPA-UNAM for the financial support of a sabbatical period at PSU to SE with a PASPA fellowship.

## Author Contributions

**Conceptualization:** Tomás López-Londoño, Roberto Iglesias-Prieto.

**Data curation:** Tomás López-Londoño.

**Formal analysis:** Tomás López-Londoño.

**Funding acquisition:** Susana Enríquez, Roberto Iglesias-Prieto.

**Investigation:** Tomás López-Londoño, Susana Enríquez.

**Methodology:** Tomás López-Londoño, Roberto Iglesias-Prieto.

**Supervision:** Susana Enríquez, Roberto Iglesias-Prieto.

**Visualization:** Tomás López-Londoño.

**Writing – original draft:** Tomás López-Londoño.

**Writing – review & editing:** Tomás López-Londoño, Susana Enríquez, Roberto Iglesias-Prieto.

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
