## [Decision Letter · Decision Letter 0]

21 Aug 2023

PONE-D-23-21541Effects of surface geometry on light exposure, photoacclimation and photosynthetic energy acquisition in zooxanthellate coralsPLOS ONE

Dear Dr. López-Londoño,

Thank you for submitting your manuscript to PLOS ONE. After careful consideration, we feel that it has merit but does not fully meet PLOS ONE’s publication criteria as it currently stands. Therefore, we invite you to submit a revised version of the manuscript that addresses the points raised during the review process. All the reviewers considered the manuscript interesting, but they provided several questions that shoud be addressed before the manuscript can be accepted (see comments below).. 

We look forward to receiving your revised manuscript.

Kind regards,

Erik Caroselli

Academic Editor

PLOS ONE

Journal Requirements:

“TLL was supported by a scholarship from the Consejo Nacional de Ciencia y Tecnología (CONACYT) from México and by Pennsylvania State University startup funds to RIP. Research funding for SE and RIP was provided by CONACYT-Mexico (Project 129880, Conv-CB-2009). A PASPA fellowship from the DGAPA-UNAM supported the visit of SE to the Biology Department at Pennsylvania State University. The funders had no role in study design, data collection and analysis, decision to publish, or preparation of the manuscript.”

5. Please update your submission to use the PLOS LaTeX template. The template and more information on our requirements for LaTeX submissions can be found at http://journals.plos.org/plosone/s/latex.

“The National Commission on Aquaculture and Fisheries 578 (CONAPESCA) for research permit (DGOPA 08606.251011.3021). We also thank the DGAPA[1]579 UNAM for the financial support of a sabbatical period at PSU to SE with a PASPA”

“TLL was supported by a scholarship from the Consejo Nacional de Ciencia y Tecnología (CONACYT) from México and by Pennsylvania State University startup funds to RIP. Research funding for SE and RIP was provided by CONACYT-Mexico (Project 129880, Conv-CB-2009). A PASPA fellowship from the DGAPA-UNAM supported the visit of SE to the Biology Department at Pennsylvania State University. The funders had no role in study design, data collection and analysis, decision to publish, or preparation of the manuscript.”

Reviewers' comments:

Reviewer's Responses to Questions

**Comments to the Author**

1. Is the manuscript technically sound, and do the data support the conclusions?

Reviewer #1: Yes

Reviewer #2: Yes

Reviewer #3: Partly

2. Has the statistical analysis been performed appropriately and rigorously? 

Reviewer #1: Yes

Reviewer #2: Yes

Reviewer #3: I Don't Know

3. Have the authors made all data underlying the findings in their manuscript fully available?

Reviewer #1: Yes

Reviewer #2: Yes

Reviewer #3: No

4. Is the manuscript presented in an intelligible fashion and written in standard English?

Reviewer #1: Yes

Reviewer #2: Yes

Reviewer #3: Yes

5. Review Comments to the Author

Reviewer #1: Review of López-Londoño

This is a nice study that explores the effects of inclination of coral tissue/skeleton surface on the ability of corals to utilize sunlight. The basic message of the paper – that the ability to use sunlight is important in shallow-water corals and this need reflects their shapes and distribution – is very well known and their results will not come as a surprise to most coral biologists. Nevertheless, this is a nice study that deserves to be published. I would package my comments as “requires modest revision”.

My major criticism of the work is that it seems to overlook the seminal work by Len Muscatine, Jim Porter, and Peter Davies (and their peers) that laid the foundations of modern understanding of coral nutrition. Critically, the symbiodinium algae fix CO2 into photosynthetically fixed carbon, a fraction of which is translocated to the animal host, where some fraction is metabolized through aerobic respiration to liberate ATP that is the source of metabolic energy that fuels chemical and mechanical work; the rest is stored as a food reserve (i.e., lipid) or lost from the coral as mucus or DOC. This paper conflates all these processes and does not recognize the true complexity of the biology at play. The reliance on the PUES acronym says it all and sows the seeds of confusion: biologically, I would argue that the algae are supplying carbon to the animal host and that only a fraction turns into energy after it has been respired. The whole heterotrophic side of the nutritional equation argues strongly for tempering the interpretation of the effects of inclination angle on the energy budgets of corals.

Other issues to address:

1. Please provide more information on the 20 fragments. Critically, it is important to know whether these are 20 host genotypes or some combination of clone mates.

2. Line 132 – specify type of sensor (not type of meter)

3. Line 159 – describe what “constant agitation” means. I really hope there were control trials completed with seawater alone.

4. Please remove all inference and interpretation from results section.

Reviewer #2: 108 – Where do these fragments come from? 20 different colonies (i.e., genotypes)? Orbicella is famous for hosting multiple symbiont species, varying from colony to colony.

115 - Why facing North? The time of year would impact the direction/aspect that the rotation should have been performed? South in the winter, North in the summer depending on the azimuth angle of the sun in the sky? Please add context by providing the dates that the experiment was conducted.

121 – Can you define reaching a steady state? The Qm values in figure 2 only appear to be steady in the 90 degree angle. The 45 degree angle could be characterised as trending upwards?

131 – Specify how the PAR sensor was positioned. I.e., matching the inclination angle with the sensor pointing in the direction of the surface normal (perpendicular?)

133 – This is an interesting approach. Can you provide a reference for this and please elaborate? What was the distance between the black panel and the sensor?

136 – Please contextualise these values. Is 06:00 dawn? 19:00 1 hr after sunset? Etc

188 – This is the section with the most limitations. Extrapolating the repair rates from a study on Porites asteroides (your ref 26) where protein synthesis inhibitors were used, to the present study, is dubious. The genetic identity and light harvesting strategies of Symbiodiniaceae among P. asteroides versus O. fav from Puerto Morelos are distinct (e.g., Hennige et al 2011 Marine Biology) and this must be caveated in this section.

315 – I would caution offering these interpretations in the results section and shift this (and similar comments in above results sections) to the discussion.

396 – Can you please characterise the components of Qm that can be derived from the equation in terms of qP versus NPQ? Does Qm explicitly tell us that the photosynthetic activity is negligible?

408 – Some discussion of this finding in the context of the many studies which have found this across depth / optical depth (e.g., Hennige et al 2008 MEPS) is warranted.

478 – Please present some discussion on how normalising O2 rates of evolution and consumption / A675 to per symbiont cell might affect your narrative. That symbiont cells were not enumerated poses strong limitations on the interpretability and as such multiple scenarios must be explored in the discussion. Both symbiont chl content and density can change across the surface of a colony or across depth / light gradients. How is this expected to alter productivity? These two differing pathways of acclimation could underpin why your inclination angle P:R rates had low correlation?

Reviewer #3: The authors explain in the photoacclimation concept, concerning their research question, by describing some of the key papers in the field. When citing papers 12-15 that are on similar subject and share similar properties, it would be good for the reader to have elaborated examples of what was found in these specific studies and how this current study takes a different angle to further increase the knowledge. The authors deployed fragments of Faveolata coral (representing flat and branching morphologies) on an underwater table in different light exposure angles. They found that at shallow depth, parts of the coral that have more direct exposure may be photoinhibited during some hours of the day while shaded parts are in chronic stress of light energy deficiency. The idea of measuring photoacclimation in different angles of exposure to light has many advantages over depth depended since it excludes all other factors as spectrum, temperature, currents, etc. The results of this study to my knowledge do not reflect the photoacclimation occur with depth gradient, as some important factors as UV-B present in 3m have a vast influence over photoacclimation.

The conclusion is interesting and can be in some degree concluded from the results. While a great work of integrating plenty of sources of data was done- Some notes and questions were raised while reading this MS:

Methods:

How was the angle of the natural colonies at 5m measured? Since not always these big corals are based on a level ground.

Line 174: "were used to characterize the temporal variation of solar energy and light-driven processes according to the inclination angle of coral samples (n = 143 days)".- were there light sensors at each inclination angle during these days that measures intensity from the different angles of the sun during the day?

PAM fluorometry is important and widely used, however a very sensitive and complicated tool. In order to extract yield value there is a need for Fv/Fm calculation of the raw data, small inaccuracy in measurements (which might happen under circumstances of diving and the need in such a stable hold of the sensor), can lead to a much bigger error after any other extrapolation of values calculated from Fv/Fm. Therefor we should be very caraful and criticizing in interpreting results derived from any further extrapolation. Also there is no added value in calculating additional parameters from the value given by the PAM. It is obvious that effective yield decreases corresponding to light. There are number of reasons for FV/Fm to decrease under high light intensity (like short term photoacclimation processes), which do not reflect solely pressure that leads to damage.

Furthermore: line 195: "The amount of photosynthetic usable energy supplied (PUES) was calculated by subtracting the estimated costs of repair (Ca) from the photosynthetic output of the zooxanthellae (Pg)". There is a difference in making conclusions based on direct measurements rather than on assumptions based on calculation and estimations. Please explain why it is important to estimate the PUES. The reader could better rely on these numbers if it is compared to other studies that could measure similar numbers and maybe on a more direct method. I would suggest sticking as close as possible to the directly measured data.

Was the excitation pressure calculated from night and day values measured at each day? Is there a measurement done prior day 1?

Yield was measured for the initial 10 days, chl estimation after 10 months, photosynthesis after 12 months. How can we assimilate any connection between the parameters? This study states that its goal is to measure short and long term photoacclimation, however how can we conclude that when the short and long term methods were different? Especially when these were in different seasons. For example: photoacclimation was significant in O2 production 12 months after the experiment started, while PSII yield after a few days according to Fig 2b and average numbers in the supplementary data shows minimal differences between treatments, although the author states that this parameter is stable.

Results:

Is there an explanation to why in the diel cycle the yield is higher in dawn than during dusk?

An important figure would be the change in effective and night PSII yield of each angle vs. acclimation days.

Line280: Do these number have units? What is the calculation?

are the calculations in Fig 5 correspond to the photosynthetic parameters of each angle?

It is very confusing that the DLI calculation is in decimal numbers, assuming that the reader notices that it is not true hour: minute calculation then they should have to make the conversion themselves.

Line 356: what are these percentages mean? How come the 90 degrees angle also spend 15% for damage if it does not attain enough light barely for compensation?

Statistics is not my strongest side, however a question rises whether the right method is used? since almost in all parameters the P value was significant while the R2 is not as high.

High chloropyll content and zooxanthellae doubling in low light locations in the colony requires high energy demand, it is known that the (line 42) "colony integration allows resource translocation from source to sink sites according to metabolic needs [5-7]." Would the authors suggest that the polyps located in high light exposure provides this energy for the high demand polyps although they provide much less energy to the cumulative budget of the colony even after the long term photoacclimation has completed?

Can the authors summaries the calculation of the daily (day + night) energy budget of each angle, considering the DLI light availability X diurnal PSII yield? And is that in compliance with the difference in P vs E?

6. PLOS authors have the option to publish the peer review history of their article (what does this mean?). If published, this will include your full peer review and any attached files.

Reviewer #1: No

Reviewer #2: No

Reviewer #3: No

---

## [Author Response · Author response to Decision Letter 0]

9 Oct 2023

A major concern raised by the reviewer 1 was related to our analysis of the photosynthetic usable energy supplied by the zooxanthellae to coral hosts (PUES). The reviewer wrote: My major criticism of the work is that it seems to overlook the seminal work by Len Muscatine, Jim Porter, and Peter Davies (and their peers) that laid the foundations of modern understanding of coral nutrition. Critically, the symbiodinium algae fix CO2 into photosynthetically fixed carbon, a fraction of which is translocated to the animal host, where some fraction is metabolized through aerobic respiration to liberate ATP that is the source of metabolic energy that fuels chemical and mechanical work... This paper conflates all these processes and does not recognize the true complexity of the biology at play... The whole heterotrophic side of the nutritional equation argues strongly for tempering the interpretation of the effects of inclination angle on the energy budgets of corals. 

Authors’ reply: We totally agree with the reviewer that coral nutrition is a complex biological process in which heterotrophy plays an essential role. Indeed, in our study we don’t question the significance of heterotrophic metabolism for corals as our aim was not the estimation of all the energy budget components. Instead, our analysis centers on assessing the relevance of the variations of the autotrophic component, estimated as photosynthetic usable energy. This variability follows a consistent and predictable pattern with respect to light availability. Other factors influencing coral energy budgets, such as heterotrophy, are more difficult to predict through our quantitative model as they are more dependent on species-specific life history strategies, and do not follow a uniform pattern with changes in light availability (reviewed by Kahng et al. 2019). We have substantially expanded the Discussion section in relation to this point to prevent such misunderstanding [lines 860-866].

In response to the reviewer's comment regarding our familiarity with prior literature, we cannot agree with his/her opinion and wish to clarify that we are well-acquainted with the seminal works of Porter, Spencer-Davies, Len Muscatine, and earlier foundational contributions by Bob Trench (1974), which laid the groundwork for our understanding of coral nutrition. In fact, we have to remark that in Muscatine et al. (1984), for instance, the calculation of the fractional contribution of photosynthetically fixed carbon translocated by the zooxanthellae to the coral host was based on the oxygen production/consumption of the holobiont, using quotients that were independent of light availability. This concept has persisted and is still widely accepted, partly due to the prevailing assumption that photoprotective mechanisms, such as non-photochemical quenching (NPQ), efficiently dissipate the excess of excitation energy absorbed without incurring in significant metabolic maintenance costs for the photosynthetic activity of the symbionts. Yet, as demonstrated in our earlier work (López-Londoño et al. 2022), light intensity has an crucial effect on the rates of photodamage and in consequence on the costs of PSII repair in the zooxanthellae, ultimately affecting the amount of photosynthates translocated to their host [lines 99-108].

Our responses to particular comments from reviewer 1:

This is a nice study that explores the effects of inclination of coral tissue/skeleton surface on the ability of corals to utilize sunlight. The basic message of the paper – that the ability to use sunlight is important in shallow-water corals and this need reflects their shapes and distribution – is very well known and their results will not come as a surprise to most coral biologists. Nevertheless, this is a nice study that deserves to be published. I would package my comments as “requires modest revision”.

Authors’ reply: We thank the reviewer for his/her positive comments and would like also to note that the effect of coral morphology on light interception and energy acquisition has remained largely descriptive and qualitative, albeit with a few notable exceptions including studies primarily focused on branching species from Pacific coral reefs, which have utilized mathematical models based on geometrical laws and proxies for assessing photosynthetic capacity (e.g., Anthony et al. 2005; Hoogenboom et al. 2008; Kaniewska et al. 2008; Kaniewska et al. 2014; Lesser et al. 2021). To the best of our knowledge, no empirical evidence is yet available in the literature that explores in detail simultaneous alterations in photoacclimation responses and photosynthetic energy acquisition, encompassing both primary productivity and associated costs of maintenance of the photosynthetic activity, in relation to the inclination angle of coral surfaces.

Please provide more information on the 20 fragments. Critically, it is important to know whether these are 20 host genotypes or some combination of clone mates.

Authors’ reply: We appreciate the reviewer’s suggestion and have updated the Methods section, clarifying that the experimental coral fragments had been collected from random colonies at a consistent depth of 5 m within La Bocana Reef. The selected donor colonies were spaced at least 5 m apart, potentially originating from different genotypes. The experimental fragments had been used for data collection with non-invasive techniques in previous projects and had remained undisturbed on a flat, horizontal surface at a constant depth of 5 m for at least one year before conducting the experiment. This ensured that all fragments had a similar photoacclimation state and potentially similar Symbiodiniaceae community composition at the beginning of the experiment [lines 179-188].

specify type of sensor (not type of meter)

Authors’ reply: We have clarified that the incident photosynthetic radiation was recorded at every inclination setting using the cosine-corrected micro-quantum sensor of the diving-PAM, previously calibrated against a reference quantum sensor (LI- 1400; LI-COR, USA) [lines 246-249].

describe what “constant agitation” means. I really hope there were control trials completed with seawater alone.

Authors’ reply: We have clarified in the text of the revised manuscript that the filtered seawater was maintained under constant agitation using magnetic stirrers, in order to facilitate a more rapid oxygen exchange across the diffusive boundary layer of the sample surface and prevent the underestimations of metabolic rates. We also indicated how we controlled for photosynthesis and respiration of micro-organisms and biofilm [lines 291-297].

Please remove all inference and interpretation from results section.

Authors’ reply: We thank the reviewer for this recommendation. We have removed all these inferences and interpretation.in the Results section of the revised manuscript.

A major concern raised by reviewer 2 was: This is the section with the most limitations [our analysis of PUES]. Extrapolating the repair rates from a study on Porites asteroides (your ref 26) where protein synthesis inhibitors were used, to the present study, is dubious. The genetic identity and light harvesting strategies of Symbiodiniaceae among P. asteroides versus O. fav from Puerto Morelos are distinct (e.g., Hennige et al 2011 Marine Biology) and this must be caveated in this section.

Authors’ reply: We appreciate the useful reference that the reviewer brings to our attention (Hennige et al. 2011); we have thoroughly reviewed and expanded our Introduction section [lines 99-108] in light of this reference. Importantly, we do not think that the findings of Hennige et al. (2011) are in contradiction to our current or previous (López-Londoño et al. 2022) analyses; on the contrary, they serve to further substantiate our conclusions. Much like the discovery by Hennige et al. that “gross photoinhibition” as a function of light availability did not exhibit significant differences between P. astreoides and O. faveolata (Figure 2a), we have also observed remarkably similar trends in the rates of PSII photodamage in experiments involving multiple coral species exposed to various intensities of both natural and artificial light, and both with and without the UV component (Figure 1, confidential data: Gómez-Campo et al., in preparation). The consistency of this pattern, even when comparing contrasting coral species and variations in the spectral composition of light, underscores the paramount influence of light intensity on the rates of photodamage and subsequent costs of PSII repair, which appear to be highly consistent among coral species and in hospite zooxanthella types. This observation aligns with the underlying principle that the diurnal recovery of the maximum PSII photochemical efficiency is driven by the balance between the rates of damage and repair to PSII (Skirving et al. 2018). In this sense, it is essential to acknowledge that our approach does not disregard the fact that various coral and/or Symbiodiniaceae species can exhibit distinct photoacclimation capabilities, enabling them to thrive in specific light environments. This subject has been the focus of extensive research in our laboratories and by other researchers over recent decades (Iglesias-Prieto and Trench 1994,1997; Hoegh-Guldberg and Jones 1999; Iglesias-Prieto et al. 2004; Enríquez et al. 2005; Warner et al. 2006; Hennige et al. 2008a; Hennige et al. 2008b; Scheufen et al. 2017a).

Furthermore, we believe that the approach used by Hennige et al. (2011) is not suitable for parameterizing the costs of PSII repair and integrating this parameter into the PUES model, as corals were exposed to three light intensities during 2-h after inhibiting the synthesis of the D1 protein with CAP, essential for PSII repair. In contrast, in our previous work we parameterized the PSII damage accumulation as a function of total light exposure during diurnal cycles (12 h). Our PUES model relies on the estimation of the costs of PSII repair as a function of the amplitude of dial oscillations of irradiance and its variation across depth/surface inclination gradients.

Our responses to particular comments from reviewer 2:

Where do these fragments come from? 20 different colonies (i.e., genotypes)? Orbicella is famous for hosting multiple symbiont species, varying from colony to colony.

Authors’ reply: We appreciate the reviewer’s comment. We have updated the Methods, clarifying that the experimental coral fragments were collected from random colonies located at a constant depth of ~5 m, potentially originating from different genotypes. The experimental fragments had remained undisturbed on a flat, horizontal surface at a constant depth of 5 m for at least one year before conducting the experiment. This ensured that all fragments had a similar photoacclimation state at the beginning of the experiment, as well as potentially similar Symbiodiniaceae community composition dominated by Symbiodinium spp., Breviolum spp., and/or Cladocopium spp. (Hennige et al. 2011; Kemp et al. 2015) [lines 179-188].

Why facing North? The time of year would impact the direction/aspect that the rotation should have been performed? South in the winter, North in the summer depending on the azimuth angle of the sun in the sky? Please add context by providing the dates that the experiment was conducted

Authors’ reply: We agree with the reviewer comment that the time of year would impact the direction/aspect that the rotation should have been performed. However, we made a deliberate decision to maintain the panel in the same position to minimize any potential disturbances that could affect the photoacclimative condition and the survivorship of our samples. Therefore, in order to prevent misinterpretations, we have clarified in the revised manuscript that the panel was oriented in a north-south direction to ensure a symmetrical diurnal cycle around noon time. The dates that the experiment was conducted were also included in the Methods section [lines 193-207].

Can you define reaching a steady state? The Qm values in figure 2 only appear to be steady in the 90 degree angle. The 45 degree angle could be characterised as trending upwards?

Authors’ reply: We appreciate the reviewer's observation. We have now updated the Methods section to clarify that the steady state of PSII photochemical activity was confirmed through the complete recovery of the maximum PSII photochemical efficiency (Fv/Fm) at each inclination setting during diurnal cycle analysis [lines 219-221]. In the Results [lines 402-406] and Discussion sections [lines 631-634], we have emphasized that corals in certain experimental settings did not achieve a steady-state Qm, which may be attributed to the gradual increases in both light exposure and temperature characteristic of the spring season. This inference is supported by a reference and an updated version of Figure 2A.

Specify how the PAR sensor was positioned. I.e., matching the inclination angle with the sensor pointing in the direction of the surface normal (perpendicular?)

Authors’ reply: We have clarified that the PAR sensor was carefully positioned perpendicular to the coral surface at each inclination setting (i.e., pointing in the direction of the normal surface) with the help of a custom-made guiding panel [lines 249-251].

This is an interesting approach. Can you provide a reference for this and please elaborate? What was the distance between the black panel and the sensor?

Authors’ reply: We have provided a reference supporting our approach and clarified that three replicates of the total incident photosynthetic radiation were recorded (Et), each one immediately followed by another measurement of the diffuse irradiance (Edf) obtained by placing a black panel at a constant distance of 10 cm above the sensor to remove the direct irradiance (Edr) [lines 252-255].

Please contextualise these values. Is 06:00 dawn? 19:00 1 hr after sunset? Etc

Authors’ reply: We have clarified that the quantum yield of PSII in the experimental corals was measured over a diurnal cycle from dawn to dusk (06:00-19:00) on a cloudless day [lines 221-223].

I would caution offering these interpretations in the results section and shift this (and similar comments in above results sections) to the discussion.

Authors’ reply: We thank the reviewer’s suggestion and have revised the Results section, removing inferences and interpretation.

Can you please characterise the components of Qm that can be derived from the equation in terms of qP versus NPQ? Does Qm explicitly tell us that the photosynthetic activity is negligible?

Authors’ reply: We are unable to characterize the components of Qm. This limitation arises from the protocol of exposing samples to brief periods of dark adaptation before each measurement in order to relax qP, as outlined in Iglesias-Prieto et al. (2004). We realized that this specific detail was absent from the Methods section, and we have since included it in the updated version of our manuscript [lines 229-242]. Furthermore, we acknowledge the reviewer's observation that low Qm values near 0 in vertically oriented corals do not explicitly indicate ‘negligible photosynthetic activity' of the symbiotic algae, but instead, they suggest that even during periods of maximum irradiance most PSII reaction centers remain open, indicating light-limited photosynthesis, as outlined by Iglesias-Prieto et al. (2004). The reliability of this parameter aligns consistently with the vertical distribution of coral species (Iglesias-Prieto et al. 2004; López-Londoño et al. 2021; Alvarez-Filip et al. 2022; Prada et al. 2022). We have revised both the Results and Discussion sections by replacing the term 'negligible photosynthetic activity' with 'low photosynthesis at the peak of irradiance' for improving text clarity.

Some discussion of this finding in the context of the many studies which have found this across depth / optical depth (e.g., Hennige et al 2008 MEPS) is warranted.

Authors’ reply: We appreciate the reviewer’s suggestion; however, we believe that placing a strong emphasis on the effect of depth/optical depth in clear vs turbid environments is beyond the scope of our study which is focused on the effect of colony geometry on light interception and energy acquisition. Nevertheless, we have expanded our discussion when addressing the complex interplay between photoacclimation responses along inclination gradients and depth gradients, following patterns that can be influenced by water optical properties, substrate type and landscape architecture [lines 958-974].

Please present some discussion on how normalising O2 rates of evolution and consumption / A675 to per symbiont cell might affect your narrative. That symbiont cells were not enumerated poses strong limitations on the interpretability and as such multiple scenarios must be explored in the discussion. Both symbiont chl content and density can change across the surface of a colony or across depth / light gradients. How is this expected to alter productivity? These two differing pathways of acclimation could underpin why your inclination angle P:R rates had low correlation?

Authors’ reply: We acknowledge the reviewer's suggestion; however, our study did not encompass the measurement of variables requiring destructive techniques, such as symbiont density/size/ID, host protein, and pigmentation. As a result, identification of specific variables underlying the P:R correlation is limited by our experimental approach and we hesitate to correlate it with specific variables. Nevertheless, we have expanded our Discussion emphasizing that different normalizations beyond surface area (e.g., to host protein, symbiont cell/ID/size, chlorophyll content), can result in different patterns of host productivity depending on the bio-optical and bio-physical characteristics of both the coral host and algal symbionts (Hennige et al. 2009; Scheufen et al. 2017a; Scheufen et al. 2017b) [lines 829-835].

Our responses to comments from reviewer 3: 

The authors explain in the photoacclimation concept, concerning their research question, by describing some of the key papers in the field. When citing papers 12-15 that are on similar subject and share similar properties, it would be good for the reader to have elaborated examples of what was found in these specific studies and how this current study takes a different angle to further increase the knowledge.

Authors’ reply: We appreciate the thoughtful comments provided by the reviewer. A brief description of what previous studies have found has now been included in the updated version of the manuscript [lines 48-53]. We have also highlighted how study provides a unique perspective to advance our comprehension of photoacclimation responses in corals as a function of the surface inclination angle and their broader implications on the energetic performance of entire colonies [lines 78-82 and 109-174].

The authors deployed fragments of Faveolata coral (representing flat and branching morphologies) on an underwater table in different light exposure angles. They found that at shallow depth, parts of the coral that have more direct exposure may be photoinhibited during some hours of the day while shaded parts are in chronic stress of light energy deficiency. The idea of measuring photoacclimation in different angles of exposure to light has many advantages over depth depended since it excludes all other factors as spectrum, temperature, currents, etc.

Authors’ reply: We appreciate the reviewer’s comment. We would like to note that O. faveolata typically forms massive colonies and thus does not represent flat or branching morphologies. However, the predominant flat morphology at the meso-scale that limits the formation of significant surface light gradients, was an essential trait of this species that allowed us to study local photoacclimation responses associated with coral surface inclination.

The results of this study to my knowledge do not reflect the photoacclimation occur with depth gradient, as some important factors as UV-B present in 3m have a vast influence over photoacclimation.

Authors’ reply: We agree with the reviewer that UV-B is an important environmental factor at 3m depth and acknowledge the variable effect that UV-B can have on photoacclimation and PSII photodamage, as outlined by, e.g., Takahashi et al. (2010). In this sense, we have expanded our discussion to briefly address this variable effect [lines 663-669]. However, it’s important to note that UV-B not only affect the photosynthetic activity by damaging PSII reaction centers, but also other cellular processes by damaging essential molecules such as proteins, lipids, and DNA. Therefore, the influence of UV-B extends beyond its effects on photosynthetic processes, encompassing broader cellular functions. In our study, we focused on the impact of PAR, as we have observed remarkably similar trends in the rates of PSII photodamage in experiments involving multiple coral species exposed to various intensities of both natural and artificial light, both with and without the UV component (Figure 1, confidential data: Gómez-Campo et al., in preparation). The consistency of this pattern, even when considering contrasting coral species and variations in the spectral composition of light, underscores the paramount influence of light intensity on the rates of photodamage and subsequent costs of PSII repair, which appear to be highly consistent among coral species and in hospite zooxanthella types. This consistency and the fact that the spectral composition of light was nearly constant in our experiments, supports that the observed photoacclimation responses and physiological performance of corals along inclination and depth gradients can be attributed exclusively to changes in total PAR, not light quality.

The conclusion is interesting and can be in some degree concluded from the results. While a great work of integrating plenty of sources of data was done- Some notes and questions were raised while reading this MS:

Authors’ reply: We appreciate the reviewer’s comments and suggestions.

How was the angle of the natural colonies at 5m measured? Since not always these big corals are based on a level ground.

Authors’ reply: In situ data were collected along transects facing north laid from top to bottom of three massive colonies of O. faveolata, controlling the inclination angle with a buoyancy device attached to a customized protractor. We have clarified that only flat coral areas with measurable, consistent inclination angles between 0� and 90� were considered in this analysis to ensure measurement accuracy [lines 227-229].

[Data of sea surface irradiance recorded by a nearby meteorological station] were used to characterize the temporal variation of solar energy and light-driven processes according to the inclination angle of coral samples (n = 143 days)".- were there light sensors at each inclination angle during these days that measures intensity from the different angles of the sun during the day?

Authors’ reply: We did not have light sensors deployed at each inclination setting. We relied on sea surface irradiance data recorded by a nearby meteorological station to assess the temporal variation of solar energy. The total downwelling irradiance at our experimental depth (3 meters), was calculated based on the Lambert-Beer law (Ez = E0 e-Kd z). Subsequently, we employed correction factors, derived from empirical measurements of local irradiance at noon, to calculate the local available irradiance at each inclination setting [lines 313-321].

PAM fluorometry is important and widely used, however a very sensitive and complicated tool. In order to extract yield value there is a need for Fv/Fm calculation of the raw data, small inaccuracy in measurements (which might happen under circumstances of diving and the need in such a stable hold of the sensor), can lead to a much bigger error after any other extrapolation of values calculated from Fv/Fm. Therefor we should be very caraful and criticizing in interpreting results derived from any further extrapolation.

Authors’ reply: We appreciate the thoughtful comments provided by the reviewer. We are well-acquainted with the sensitivity and intricacies of the PAM fluorometry technique, given our pioneering work in utilizing this technique on several marine primary producers and our extensive contributions to this field through numerous academic papers. In our study, we used standard protocols and controlled experimental conditions aimed at minimizing potential noise in our analyses.

Also there is no added value in calculating additional parameters from the value given by the PAM. It is obvious that effective yield decreases corresponding to light. There are number of reasons for FV/Fm to decrease under high light intensity (like short term photoacclimation processes), which do not reflect solely pressure that leads to damage.

Authors’ reply: We respectfully disagree with the reviewer’s suggestion regarding the value of calculating additional parameters from the PAM data. Qm serves as a useful proxy for assessing the physiological performance and the excessive energy absorbed and dissipated as heat (NPQ), already used and tested in several published studies (e.g., Iglesias-Prieto et al. 2004; Warner et al. 2006; Warner et al. 2010; López-Londoño et al. 2021; Prada et al. 2022). Its simplicity allows for reliable indications of whether photosynthetic rates are within the range of acute light-limitation (values close to 0) and near full saturation and potential photoinhibition (values close to 1), particularly when calculated as Qm at the diurnal peak of irradiance. This information is not so clearly derived from the values of Fv/Fm or the diurnal oscillations of �F/Fm’, independent of their clear connection with light availability. Furthermore, the reliability of this parameter aligns consistently with the vertical distribution of coral species (Iglesias-Prieto et al. 2004; López-Londoño et al. 2021; Prada et al. 2022). Either in this study or in previous ones, we never assume that the diurnal oscillation of Fv/Fm is exclusively linked to photodamage.

The amount of photosynthetic usable energy supplied (PUES) was calculated by subtracting the estimated costs of repair (Ca) from the photosynthetic output of the zooxanthellae (Pg). There is a difference in making conclusions based on direct measurements rather than on assumptions based on calculation and estimations. Please explain why it is important to estimate the PUES. The reader could better rely on these numbers if it is compared to other studies that could measure similar numbers and maybe on a more direct method. I would suggest sticking as close as possible to the directly measured data.

Authors’ reply: We appreciate the thoughtful feedback provided by the reviewer. While we did not directly quantify the photosynthetic usable energy supplied (PUES), our approach for estimating this parameter relies on robust physiological evidence derived from direct characterizations [lines 99-108]. Direct measurements of the energy expenditure of the zooxanthellae linked to the repair of PSII from photodamage as a function of light exposure are practically non-existent in the literature. Research partially connected to this subject has primarily resorted to bulk stable isotopes for assessing coral trophic ecology and the relative dependence on photosynthesis with respect to heterotrophy (e.g., Houlbrèque and Ferrier-Pagès 2009; Conti-Jerpe et al. 2020; Lesser et al. 2022). Other studies have documented the role of non-photochemical quenching to protect the photosynthetic apparatus of the zooxanthellae and optimize energetic performance (e.g., Brown et al. 1999; Hoegh-Guldberg and Jones 1999; Lesser and Gorbunov 2001; Hennige et al. 2008a; Warner et al. 2010; Suggett et al. 2011). To our knowledge, the study by López-Londoño et al. (2022) likely stands as one of the pioneering efforts in this regard.

Was the excitation pressure calculated from night and day values measured at each day? Is there a measurement done prior day 1?

Authors’ reply: The maximum excitation pressure over PSII (Qm) was calculated based on measurements of the maximum (Fv/Fm) and the effective (ΔF/Fm’) quantum yields of photosystem II (PSII) respectively recorded at dusk and at local noon on the same days. We have updated the Results section, including in the Figure 2A the measurements of Qm recorded during four days before adjusting the inclination angles.

Yield was measured for the initial 10 days, chl estimation after 10 months, photosynthesis after 12 months. How can we assimilate any connection between the parameters? This study states that its goal is to measure short and long term photoacclimation, however how can we conclude that when the short and long term methods were different? Especially when these were in different seasons. For example: photoacclimation was significant in O2 production 12 months after the experiment started, while PSII yield after a few days according to Fig 2b and average numbers in the supplementary data shows minimal differences between treatments, although the author states that this parameter is stable.

Authors’ reply: We appreciate the thoughtful comments provided by the reviewer. It's important to note that our primary objective was to assess the correlation between short-term (e.g., quantum yield of PSII) and long-term (e.g., pigmentation, photosynthetic parameters) photoacclimation responses in relation to the inclination angle of coral surface. This was more important than exploring potential correlations among physiological parameters, considering the inherent time differences involved in these measurements. To further clarify our experimental approach, we would like to remark that:

- The consistency between the patterns of the quantum yield of PSII obtained in experimental samples and in coral colonies within their native environments suggests that the short-term photoacclimation responses induced by adjusting the inclination angle of coral surfaces, likely reflect natural processes occurring in coral colonies [lines 659-663].

- All measurements were carried out under similar seasonal conditions, with most physiological parameters assessed during the spring of 2014 and 2015, and the optical properties of the coral tissue in late winter of 2015. Our study did not aim to investigate seasonal variations in physiological parameters but differences among the photoaclimatory conditions induced by changes in the inclination angle of coral surfaces.

- Given that we conducted measurements for each physiological parameter with minimal time intervals (e.g., same day for the quantum yield of PSII, within a few days for the photosynthetic parameters derived from PE curves), our comparative analysis of each photoacclimation response along the inclination gradient is well-founded.

- The apparent lack of a steady-state Qm in some experimental settings may be attributed to the gradual increases in both light exposure and temperature characteristic of the spring season [lines 631-634].

Is there an explanation to why in the diel cycle the yield is higher in dawn than during dusk?

Author’s reply: We have now updated the Results and Discussion sections, clarifying that corals in certain experimental settings did not achieve a steady-state Qm, which may be attributed to the gradual increases in both light exposure and temperature characteristic of the spring season [lines 631-634]. This natural, uncontrollable variations in environmental conditions could account for the slightly higher values of Fv/Fm at dusk relative to dawn in some inclination settings. This inference is supported by our published findings referenced and the updated Figure 2A.

An important figure would be the change in effective and night PSII yield of each angle vs. acclimation days.

Authors’ reply: We appreciate the reviewer’s suggestion. We feel that including the figure depicting the changes in Qm clearly illustrates the effect of coral surface inclination on the PSII excitation pressure under maximum irradiance. It’s important to highlight that this parameter has already been used and tested in several published studies (Iglesias-Prieto et al. 2004; Warner et al. 2006; Warner et al. 2010; López-Londoño et al. 2021; Prada et al. 2022). Its simplicity allows for reliable indications of whether photosynthetic rates are within the range of acute light-limitation (values close to 0) or near full saturation and potential photoinhibition (values close to 1) at the diurnal peak of irradiance (noon). This information is not so clearly derived from the values of Fv/Fm or the diurnal oscillations of �F/Fm’, independent of their clear connection with light availability. Furthermore, the reliability of this parameter aligns consistently with the vertical distribution of coral species (Iglesias-Prieto et al. 2004; López-Londoño et al. 2021; Prada et al. 2022).

Do these number have units? What is the calculation? are the calculations in Fig 5 correspond to the photosynthetic parameters of each angle?

Authors’ reply: Values of Absorptance at 675nm, A675, have no units as this parameter represents the relative fraction (or percentage) of the amount of incident light absorbed (EA/EI). The unitless nature of this parameter is indicated in the Supplementary Table. The Figure 5 is derived from analysis using the mean values of the P vs E curve parameters at each inclination setting. This information has been included in the Methods section [lines 333-334].

It is very confusing that the DLI calculation is in decimal numbers, assuming that the reader notices that it is not true hour: minute calculation then they should have to make the conversion themselves.

Authors’ reply: The units of the daily light integral (DLI) are mol quanta m-2 d-1. There was an error in the absolute values reported in Figure 5B, and appreciate the reviewer’s comment, which allowed us correcting it in the updated version.

what are these percentages mean? How come the 90 degrees angle also spend 15% for damage if it does not attain enough light barely for compensation?

Authors’ reply: The costs of repair from photodamage (Ca) are calculated in percentage of the total amount of photosynthetically fixed energy (Pg), as has been indicated in the Methods section [lines 338-340]. The rates of PSII photodamage exhibit a non-linear relationship with light exposure (López-Londoño et al. 2022). Consequently, even exposure to low light intensities (e.g., below the compensating irradiance, Ec) results in minimal photodamage to PSII reaction centers, incurring associated repair costs.

Statistics is not my strongest side, however a question rises whether the right method is used? since almost in all parameters the P value was significant while the R2 is not as high.

Authors’ reply: The linear regression models were used for estimating the relationships between dependent variable (physiological parameters) and the independent variable (inclination angle of coral surface). The p-value and R2 offer distinct insights into the performance of these linear models: the p-values inform the model’s significance level, while the R2 inform how well the model fits the data, essentially measuring the proportion of variation in the dependent variable explained by the independent variable. It's essential to note that there is no predetermined relationship between p-value and R2.

High chloropyll content and zooxanthellae doubling in low light locations in the colony requires high energy demand, it is known that the (line 42) "colony integration allows resource translocation from source to sink sites according to metabolic needs [5-7]." Would the authors suggest that the polyps located in high light exposure provides this energy for the high demand polyps although they provide much less energy to the cumulative budget of the colony even after the long term photoacclimation has completed?

Authors’ reply: Our data suggest that when considering both primary productivity (Pg) and the cost of repair from photodamage (Ca), polyps situated in high light-exposed areas of the colony, such as horizontal surfaces, may not necessarily be the most productive as a result of the increased energy costs of repair from photodamage in the zooxanthellae. Consequently, we propose that the greater extension and calcification commonly observed in these areas might result from the importation of energy products from other, more productive source sites within the colony, where the balance between Pg and Ca yields maximum energetic output (e.g., in coral surfaces with an inclination angle between 25° and 45°) [lines 942-953].

Can the authors summaries the calculation of the daily (day + night) energy budget of each angle, considering the DLI light availability X diurnal PSII yield? And is that in compliance with the difference in P vs E?

Authors’ reply: Our energy budget analyses were centered on light-driven processes. Given that we did not observe a significant relationship between respiration rates and the inclination angle of the coral surface, we cannot definitively assert that metabolic processes during the night impact in a different way the energy budget of corals according to the inclination angle. The relationships of the DLI with the maximum quantum yield of PSII (Fv/Fm) and other physiological parameters can be inferred from figures 2B-2D and 3B (insert).

---

## [Decision Letter · Decision Letter 1]

8 Nov 2023

PONE-D-23-21541R1Effects of surface geometry on light exposure, photoacclimation and photosynthetic energy acquisition in zooxanthellate coralsPLOS ONE

Dear Dr. López-Londoño,

Thank you for submitting your manuscript to PLOS ONE. After careful consideration, we feel that it has merit but does not fully meet PLOS ONE’s publication criteria as it currently stands. Therefore, we invite you to submit a revised version of the manuscript that addresses the points raised during the review process. Two of the three reviewers have endorsed publicatoin of the manuscript. Please provide a revised versionafter considering the last comments by Reviewer 3.  Please submit your revised manuscript by Dec 23 2023 11:59PM. If you will need more time than this to complete your revisions, please reply to this message or contact the journal office at plosone@plos.org. Please include the following items when submitting your revised manuscript:A rebuttal letter that responds to each point raised by the academic editor and reviewer(s). You should upload this letter as a separate file labeled 'Response to Reviewers'.A marked-up copy of your manuscript that highlights changes made to the original version. You should upload this as a separate file labeled 'Revised Manuscript with Track Changes'.An unmarked version of your revised paper without tracked changes. You should upload this as a separate file labeled 'Manuscript'.If applicable, we recommend that you deposit your laboratory protocols in protocols.io to enhance the reproducibility of your results. Protocols.io assigns your protocol its own identifier (DOI) so that it can be cited independently in the future. For instructions see: https://journals.plos.org/plosone/s/submission-guidelines#loc-laboratory-protocols. Additionally, PLOS ONE offers an option for publishing peer-reviewed Lab Protocol articles, which describe protocols hosted on protocols.io. Read more information on sharing protocols at https://plos.org/protocols?utm_medium=editorial-email&utm_source=authorletters&utm_campaign=protocols.

We look forward to receiving your revised manuscript.

Kind regards,

Erik Caroselli

Academic Editor

PLOS ONE

Journal Requirements:

Reviewers' comments:

Reviewer's Responses to Questions

**Comments to the Author**

1. If the authors have adequately addressed your comments raised in a previous round of review and you feel that this manuscript is now acceptable for publication, you may indicate that here to bypass the “Comments to the Author” section, enter your conflict of interest statement in the “Confidential to Editor” section, and submit your "Accept" recommendation.

Reviewer #1: All comments have been addressed

Reviewer #2: All comments have been addressed

Reviewer #3: All comments have been addressed

2. Is the manuscript technically sound, and do the data support the conclusions?

Reviewer #1: Yes

Reviewer #2: Yes

Reviewer #3: Yes

3. Has the statistical analysis been performed appropriately and rigorously? 

Reviewer #1: Yes

Reviewer #2: Yes

Reviewer #3: Yes

4. Have the authors made all data underlying the findings in their manuscript fully available?

Reviewer #1: Yes

Reviewer #2: Yes

Reviewer #3: Yes

5. Is the manuscript presented in an intelligible fashion and written in standard English?

Reviewer #1: Yes

Reviewer #2: Yes

Reviewer #3: Yes

6. Review Comments to the Author

Reviewer #1: I liked the first submission and had only small suggested. These have been addressed very well. I am happy to recommend publication.

Reviewer #2: I am satisfied with the responses of the authors and thank them for addressing my concerns. I recommend that this article progress to publication.

Reviewer #3: the renovated manuscript is much more cleare and presents the idea and provides enough supporting data. however some minor issues are still raised in the attached file.

7. PLOS authors have the option to publish the peer review history of their article (what does this mean?). If published, this will include your full peer review and any attached files.

Reviewer #1: No

Reviewer #2: No

Reviewer #3: No

---

## [Author Response · Author response to Decision Letter 1]

13 Nov 2023

The reviewer wrote: The conclusion of this paper is rather simple and important. In order to derive that, it is enough to understand only part of the results. I question some methods/calculations used here, some were used for the first time with no reference or direct measurement to assure its accuracy and some calculations and extrapolations of the data are known from previous studies but are not significant for the purpose of this study.

The authors should think carefully if using these data will strength or weakens the study, especially when the title of this study mentions photoacclimation and photosynthesis, and the derivations like PUES and Ca are not central for the final conclusion.

Authors’ reply: We appreciate the reviewer's acknowledgment of the key conclusion in our paper. Regarding the suggestion that only a portion of the results adequately supports our conclusions, and that including the PUES analysis may weaken our study as it is not significant for the purpose of this study, we appreciate the reviewer's opinion but respectfully disagree. Our study comprises three integral components—changes in light exposure, photoacclimation, and photosynthetic energy acquisition with the inclination angle of coral surfaces, each supported by corresponding sections in the Methods and Results. It's essential to note that changes in photosynthetic energy acquisition cannot be reliably inferred solely from alterations in light exposure and photoacclimation status without considering the energy costs of repair from photodamage for the in hospite zooxanthellae. In our previously published work (López-Londoño et al., 2022), we detailed the changes in photodamage rates and the associated costs of photorepair in relation to varying light exposure levels. While we recognize that an in-depth examination of the intricate dynamics of PSII photodamage and repair processes in response to light exposure extends beyond the scope of our current study, it is important to note that the fundamental patterns have already been documented in our earlier publication. Furthermore, we are currently preparing a new manuscript that delves into a comprehensive analysis of the rates of photodamage and the related costs of photorepair in symbiotic zooxanthellae, expanding upon our previous findings and exploring these processes in greater depth.

Concerning the inquiry about certain methods/calculations used in our study to estimate the relative change in PUES, we have revised the Methods section to offer a more comprehensive description of the equations and values employed in this analysis, along with the rationale for selecting these values [lines 230-254].

The reviewer wrote: The title of this study should be more specific to the coral used for this study Orbicella faveolata, and not zooxanthellae corals? 

Authors’ reply: We value the reviewer's suggestion but respectfully disagree. The light interception at a specific colony surface area is governed by geometrical relations, which exert a consistent influence on all corals and even non-living surfaces. Our primary objective in this analysis is to depict broader patterns of change in photoacclimation and energetic performance that are dictated by these geometrical relations, rather than confining the study to providing species-specific absolute values.

The reviewer wrote: This study concludes that colony geometry plays an essential role in shaping the energy balance. what is missing here is the deduction of this conclusion to the actual morphologies in nature. If these corals form vast colonies, then what percentage of the colony suffers from excessive irradiance and light-limiting conditions according to its angles?

Authors’ reply: We appreciate the insightful observation provided by the reviewer. We concur with the reviewer's acknowledgment that the percentage of the colony surface at a specific inclination angle significantly influences the energetic performance of the entire colony within a given light climate along the depth gradient. However, our study's conclusion is that "colony geometry determines light availability, holobiont physiology, and photosynthetic energy acquisition in zooxanthellate corals”. In line with this, we propose that "studying the entire coral colony and its metabolic gradients should be a research priority to enhance our understanding of colonial responses to environmental cues and the role of colony morphology and energetic economy on the vertical distribution of species and niche diversification in coral reefs." This direction not only aligns with our findings but also resonates with the comment made by the reviewer, underscoring its relevance and potential impact in the field.

---

## [Editor Report · Decision Letter 2]

20 Nov 2023

Effects of surface geometry on light exposure, photoacclimation and photosynthetic energy acquisition in zooxanthellate corals

PONE-D-23-21541R2

Dear Dr. López-Londoño,

We’re pleased to inform you that your manuscript has been judged scientifically suitable for publication and will be formally accepted for publication once it meets all outstanding technical requirements.

Kind regards,

Erik Caroselli

Academic Editor

PLOS ONE
---

## [Editor Report · Acceptance letter]

8 Dec 2023

PONE-D-23-21541R2 

Effects of surface geometry on light exposure, photoacclimation and photosynthetic energy acquisition in zooxanthellate corals 

Dear Dr. López-Londoño:

I'm pleased to inform you that your manuscript has been deemed suitable for publication in PLOS ONE. Congratulations! Your manuscript is now with our production department. 

Kind regards, 

on behalf of

Dr. Erik Caroselli 

Academic Editor

PLOS ONE